# Marcksl1 modulates endothelial cell mechanoresponse to haemodynamic forces to control blood vessel shape and size

Igor Kondrychyn [1], Douglas J. Kelly[1], Núria Taberner Carretero [1], Akane Nomori[1], Kagayaki Kato [2,3], Jeronica Chong[1], Hiroyuki Nakajima [4], Satoru Okuda [5], Naoki Mochizuki [4] & Li-Kun Phng [1✉]

The formation of vascular tubes is driven by extensive changes in endothelial cell (EC) shape. Here, we have identified a role of the actin-binding protein, Marcksl1, in modulating the mechanical properties of EC cortex to regulate cell shape and vessel structure during angiogenesis. Increasing and depleting Marcksl1 expression level in vivo results in an increase and decrease, respectively, in EC size and the diameter of microvessels. Furthermore, endothelial overexpression of Marcksl1 induces ectopic blebbing on both apical and basal membranes, during and after lumen formation, that is suppressed by reduced blood flow. High resolution imaging reveals that Marcksl1 promotes the formation of linear actin bundles and decreases actin density at the EC cortex. Our findings demonstrate that a balanced network of linear and branched actin at the EC cortex is essential in conferring cortical integrity to resist the deforming forces of blood flow to regulate vessel structure.

[1] Laboratory for Vascular Morphogenesis, RIKEN Center for Biosystems Dynamics Research, Kobe 650-0047, Japan. [2] Bioimage Informatics Group, Exploratory Research Center on Life and Living Systems (ExCELLS), National Institutes of Natural Sciences, Okazaki, Aichi 444-8585, Japan. [3] Laboratory of Biological Diversity, National Institute for Basic Biology, National Institutes of Natural Sciences, Okazaki, Aichi 444-8585, Japan. [4] Department of Cell Biology and AMED-CREST, National Cerebral and Cardiovascular Center, Research Institute, Suita, Osaka 565-8565, Japan. [5] WPI Nano Life Science Institute, Kanazawa University, Kanazawa 920-1192, Japan. ✉email: likun.phng@riken.jp

The efficient distribution and exchange of gases and solutes to cells and tissues at distal portions of the body are facilitated by tubular organs. In vertebrates, oxygen and metabolites are distributed throughout the body by blood flow through vascular tubes. The tubulogenesis, or luminenisation, of blood vessels is a multistep process that requires the establishment of apical-basal polarity, the remodeling of cell-cell adhesion and cell shape changes[1]. In the dorsal aorta, which is the first vessel that forms within the developing embryo, luminenisation is initiated by the clearance of adhesion complexes such as vascular endothelial (VE)-cadherin at the intercellular junction, from the center to the periphery, of two opposing angioblasts and the separation of the pre-apical membranes[2,3]. The clearance of adhesion complexes requires localised actomyosin activity that is regulated by Moesin and the Rasip1-Arhgap29-Cdc42-Pak4 pathway[2–4] while the separation of the pre-apical membranes is mediated by electrostatic repulsion of negatively charged apical CD34 sialomucins such as Podocalyxin[5]. In addition, the balance between EC-EC and EC-ECM adhesion is critical for lumen formation[4,6]. These cellular processes consequently remodel the shape of the apical membrane and flatten the ECs to open a central lumen. Similarly, the remodeling of intercellular junctions, EC rearrangement and cell shape changes drive luminenisation of angiogenic sprouts in a process called cord hollowing[7].

In the processes described above, ECs generate intrinsic forces to drive cell shape changes necessary for vascular tube formation. However, at later stages of development after blood circulation is established, many angiogenic vessels are exposed to the physical forces that arise from blood flow (haemodynamic forces) during tubulogenesis. These extrinsic forces in turn contribute to lumen expansion in endothelial tip cells in a process termed transcellular invagination[7] and are critical in keeping lumens open[8]. High spatiotemporal imaging revealed that blood pressure deforms the apical membrane to generate inverse blebs, which are spherical protrusions that expand into the cytoplasm as a result of a weak apical membrane-to-cortex attachment and a high pressure gradient between the lumen and the cell cytoplasm[9]. Inverse blebs are highly dynamic structures that require tight regulation since excessive blebbing is detrimental to lumen formation. ECs control these membrane blebs by triggering a repair mechanism consisting of local and transient actomyosin activity around the bleb cortex that drives bleb retraction, normalisation of apical membrane and unidirectional lumen expansion in the vessel[9]. The strength of haemodynamic forces and how ECs respond to these forces are therefore critical in regulating lumen formation through transcellular invagination and the final size of the lumen. Apart from one study that showed that the suppression of non-muscle myosin II activity, and hence stiffness, by Rasip1 at the apical membrane is required for lumen expansion during embryonic growth[3], little else is known about how ECs molecularly regulate lumen expansion.

In this study, we sought to understand whether and how the actin cytoskeleton regulates lumen expansion. By performing live imaging at different stages of lumen development, we detect a spatiotemporal pattern of apical actomyosin localisation that coincides with lumen expansion. Furthermore, we find that Myristoylated alanine-rich C-kinase substrate like 1 (MARCKSL1, also known as MARCKS-related protein, MRP), an actin-bundling protein[10,11], is highly enriched at the EC apical membrane and that its expression level perturbs EC size and blood vessel diameter. Additionally, the overexpression of Marcksl1 specifically in ECs causes membrane blebbing in perfused vessels. By performing high-resolution imaging, we find that Marcksl1 reorganises the actin cytoskeleton at the EC cortex by promoting the formation of linear actin bundles and decreasing actin density to weaken cortical integrity. Together,

these findings demonstrate that vessel shape and size are regulated by the balance between haemodynamic forces and resistive forces generated by ECs, and that Marcksl1 modifies EC resistance by regulating cortical actin organisation.

## Results

**Lumen expansion coincides with decreased actomyosin at the apical cortex.** Visualisation of actin and non-muscle myosin II assembly in *Tg(fli1:Lifeact-mCherry)*[ncv7] and *Tg(fli1ep:myl9b-EGFP)*[rk25] embryos, respectively, revealed differences in cortical actomyosin assembly in ECs at distinct phases of vessel formation. Using *Tg(fli1ep:EGFP-PLC1δPH)*[rk26] and *Tg(kdr-l:ras-mCherry)*[s916] to visualize the apical membrane of ECs, we detected a gradient of actomyosin network along the apical cortex during lumen expansion of intersegmental vessels (ISVs) in 1 day post-fertilisation (dpf) embryos. While there is very little or no Lifeact and Myl9b at the invaginating (anterior) front of the lumen, a higher level is observed at the posterior segment of the expanding lumen (Fig. 1a, d, g). In contrast, Lifeact (Fig. 1b, c) and Myl9b (Fig. 1e, f) are observed at both the apical and basal cortices of ECs in perfused ISVs of 2 and 3 dpf embryos, with prominent levels detected at the apical cortex. These observations suggest the existence of a temporal switch of actomyosin assembly at the apical cortex that allows lumen expansion at low levels, such as the anterior of the lumen during its formation, but confers cortical stiffness to the EC at higher levels in perfused blood vessels.

During a search for actin-binding proteins with potential roles in regulating EC behaviour, we discovered that the localisation of Marcksl1 is enriched in the apical membrane during lumen expansion. In zebrafish, two Marcksl1 paralogues, *marcksl1a* and *marcksl1b*, exist. In situ hybridisation shows that although both genes are strongly expressed in the central nervous system, they are also expressed in ECs (Supplementary Fig. 1a). Transcriptome analysis of single ECs isolated from 1 and 3 dpf embryos revealed that more ECs express *marcksl1b* than *marcksl1a* (Supplementary Fig. 1b) and that ECs express higher number of *marcksl1b* transcripts than *marcksl1a* (Supplementary Fig. 1c). By tagging Marcksl1a (Fig. 1h) or Marcksl1b (Fig. 1i) with EGFP and expressing the transgenes in a mosaic manner under the endothelial *fli1ep* promoter, we detected their localisation at the plasma membrane including filopodia during ISV formation. Notably, when luminenisation begins, there is an enrichment of both proteins in the apical, but not basal, membrane, suggesting a potential role of Marcksl1a and Marcksl1b in lumen expansion.

**Marcksl1 regulates lumen formation and blood vessel diameter.** During the mosaic analysis of ECs overexpressing either Marcksl1a or Marcksl1b, we frequently observed that these cells are wider or bulbous in appearance compared with neighboring wildtype ECs at 2 dpf. Quantification revealed that the diameters of arterial ISVs (aISVs), venous ISVs (vISVs) and dorsal longitudinal anastomotic vessel (DLAV) composed of ECs with exogenous Marcksl1a (Fig. 2b, c) or Marcksl1b (Fig. 2e, f) expression were significantly increased when compared to vessels composed of wildtype ECs. The degree of vessel dilation was potentiated when a higher level of transgene expression was induced by using the *fli1*-driven Gal4-UAS system compared to *fli1ep* promoter (Supplementary Fig. 2a, c, e, f and h).

The binding of Marcksl1 to F-actin depends on its Effector Domain (ED), a highly basic segment composed of 24-25 amino acid residues that polymerises monomeric actin and causes the alignment of F-actin into bundle-like structures[11], and JNK phosphorylation, which enables Marcksl1 to bundle and stabilize F-actin[10]. When we expressed Marcksl1a or Marcksl1b lacking

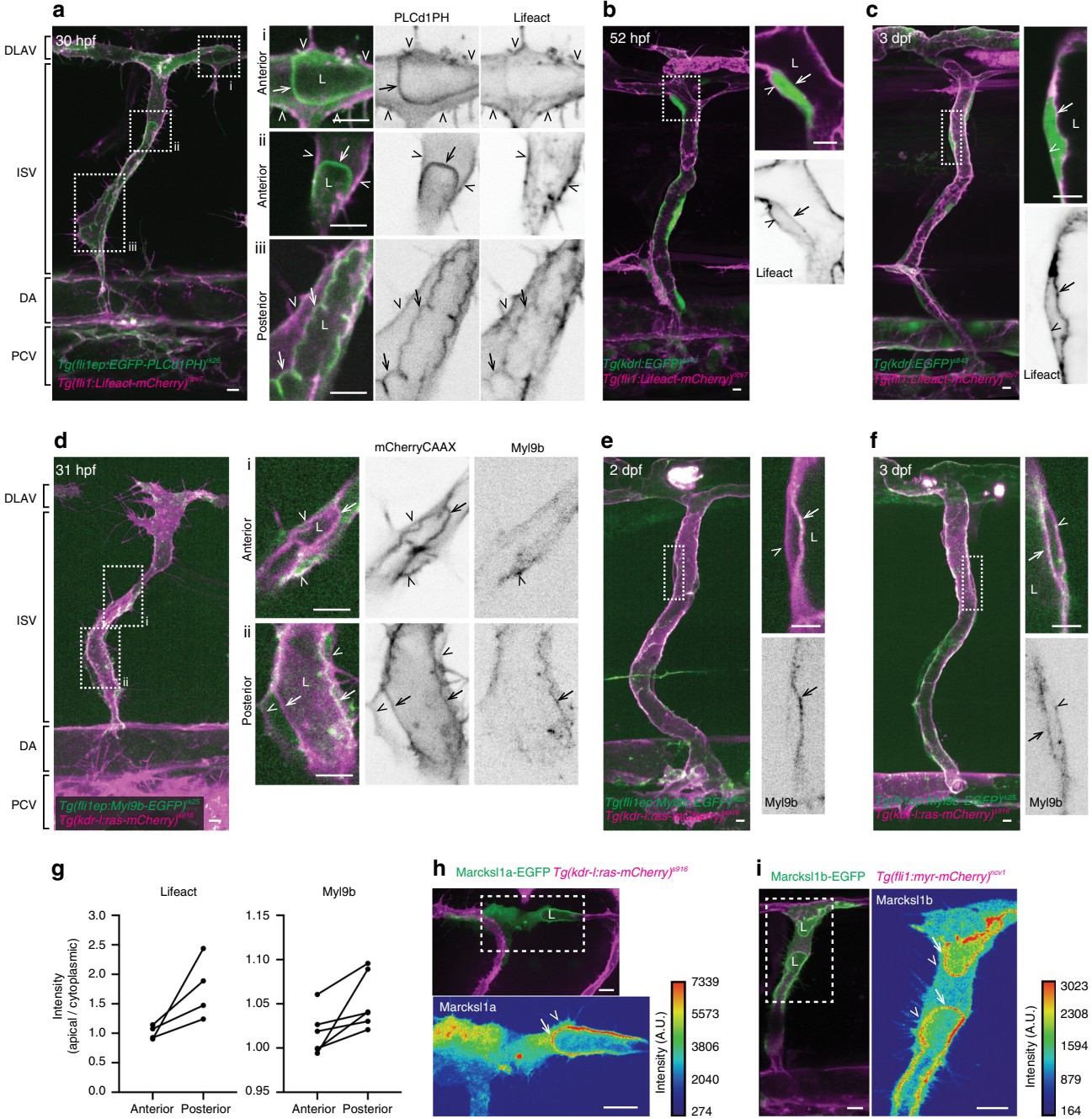

**Fig. 1 Low actomyosin at endothelial cell apical cortex coincides with lumen expansion. a–f** Maximum intensity projection of confocal z-stacks of trunk vessels at different stages of zebrafish development. Cropped images are single-plane images of the z-stack. During lumen expansion of ISVs from 30 to 34 hpf embryos, higher levels of actin (**a**, Lifeact) and non-muscle myosin II (**d**, Myl9b) are assembled at the apical cortex of the posterior region of the lumen (iii in **a**, ii in **d**) compared to the expanding anterior region of the lumen (i and ii in **a**, i in **d**), which contains very little or no actomyosin. At 2 and 3 dpf, distinct actin (**b**, **c**) and non-muscle myosin II (**e**, **f**) are detected in the apical cortex of fully lumenised vessels. Images are representative of 6 (**a**, n = 2 independent experiments), 9 (**b**, n = 5 independent experiments), 7 (**c**, n = 4 independent experiments), 23 (**d**, n = 3 independent experiments), 8 (**e**, n = 3 independent experiments) and 4 (**f**, n = 2 independent experiments) embryos. **g** Quantification of apical Lifeact and Myl9b intensity levels in the anterior and posterior segments of the same lumen at 30–34 hpf. Each data pair represents one ISV (Lifeact: n = 4 ISVs from 4 embryos; Myl9b: n = 6 ISVs from 5 embryos). **h, i** Single-cell expression of Marcksl1a-EGFP in 34 hpf *Tg(kdr-l:ras-mCherry)^s916* embryo (**h**, apical enrichment was observed in 5 out of 5 embryos from 3 independent experiments) and Marcksl1b-EGFP in 38 hpf *Tg(fli1:myr-mCherry)^ncv1* embryo (**i**, apical enrichment was observed in 20 out of 20 embryos from 6 independent experiments). Arrows, apical cortex; arrowheads, basal cortex; dashed boxes, the magnified regions; DA dorsal aorta; DLAV dorsal longitudinal anastomotic vessel; ISV intersegmental vessel; L lumen; PCV posterior cardinal vein. Scale bars, 5 μm (**a–f**) and 10 μm (**h, i**). Source data are provided as a Source data file.

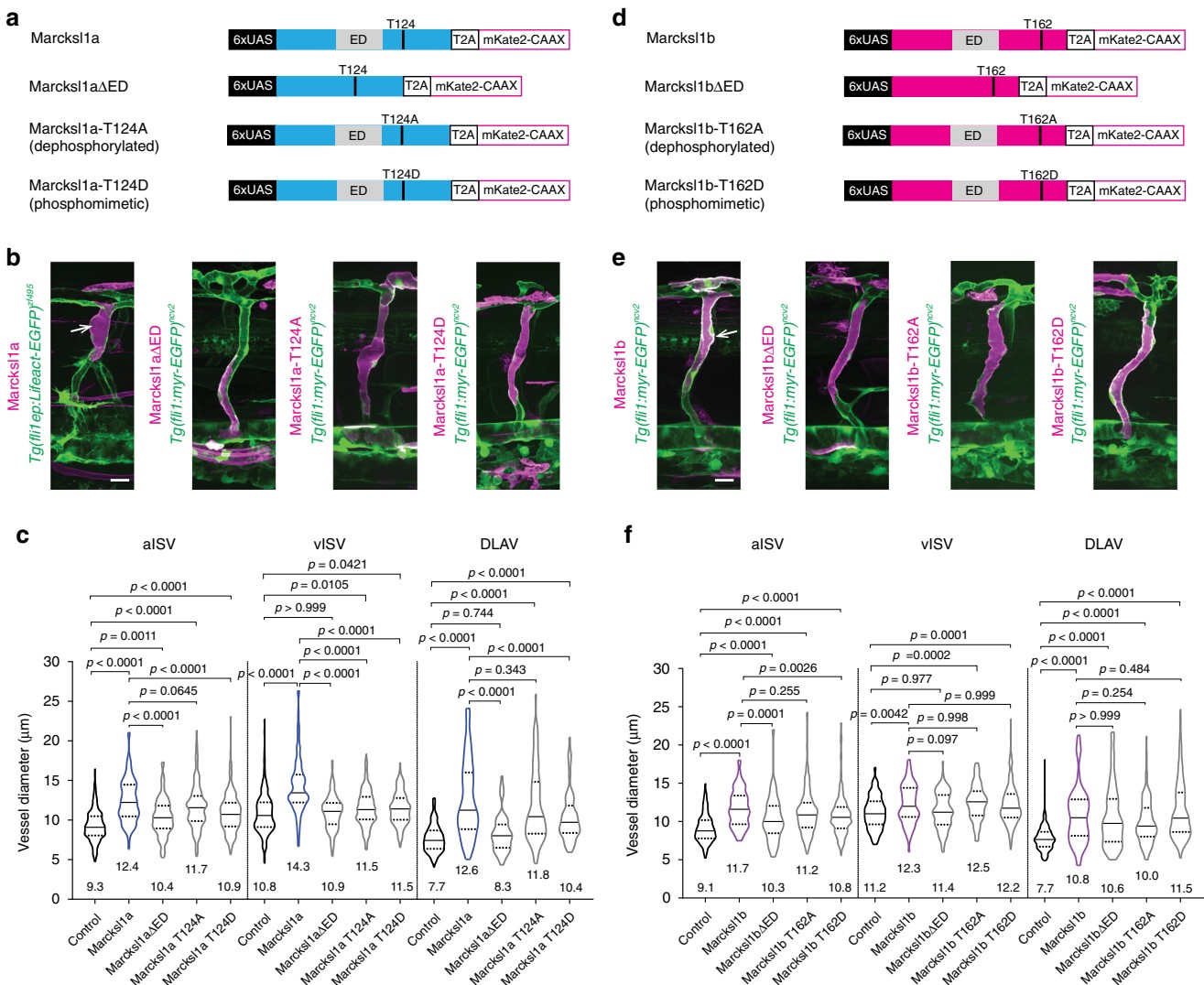

**Fig. 2 Elevated Marcksl1 expression level increases blood vessel diameter. a, d** Plasmid constructs encoding GAL4FF/*UAS*-driven expression of wildtype or mutated Marcksl1a (**a**) and Marcksl1b (**d**) proteins trailed by a self-cleaving T2A peptide fused to mKate2-CAAX. **b, e** Maximum intensity projection of confocal z-stacks of ISVs of 2 dpf *Tg(fli1:GAL4FF)*[ubs3];*Tg(fli1ep:Lifeact-EGFP)*[zf495] or *Tg(fli1:GAL4FF)*[ubs3];*Tg(fli1:myr-EGFP)*[ncv2] embryos expressing constructs encoding wildtype or mutated Marcksl1a (**b**) and Marcksl1b (**e**). Marcksl1 overexpressing cells are in magenta (cells expressing mKate2-CAAX). Arrows indicate dilated vessel. **c** Quantification of aISV, vISV and DLAV diameter in control and Marcksl1a-overexpressing blood vessels (control: $n = 60$ aISVs/51 vISVs/54 DLAVs from 85 embryos; wildtype Marcksl1a: $n = 17$ aISVs/16 vISVs/16 DLAVs from 25 embryos; Marcksl1aΔED: $n = 16$ aISVs/20 vISVs/16 DLAVs from 24 embryos; Marcksl1a-T124A: $n = 43$ aISVs/44 vISVs/35 DLAVs from 33 embryos; Marcksl1a-T124D: $n = 43$ aISVs/32 vISVs/24 DLAVs from 29 embryos). **f** Quantification of aISV, vISV and DLAV diameter in control and Marcksl1b-overexpressing vessels (control: $n = 62$ aISVs/35 vISVs/67 DLAVs from 47 embryos; wildtype Marcksl1b: $n = 23$ aISVs/14 vISVs/20 DLAVs from 24 embryos; Marcksl1bΔED: $n = 21$ aISVs/22 vISVs/25 DLAVs from 23 embryos; Marcksl1b-T162A: $n = 31$ aISVs/15 vISVs/22 DLAVs from 21 embryos; Marcksl1b-T162D: $n = 54$ aISVs/42 vISVs/34 DLAVs from 34 embryos). Violin plots represent the entire range of values, dotted lines indicate first and third quartiles, center lines are median. Mean values are indicated. Data are collected from 2 (Marcksl1bΔED), 3 (Marcksl1a-T124A, Marcksl1a-T124D, Marcksl1b-T162A and Marcksl1b-T162D) and 4 (Marcksl1aΔED, wildtype Marcksl1a and Marcksl1b) independent experiments and analyzed by ordinary one-way ANOVA with Sidak's multiple comparisons test. ED effector domain; DLAV dorsal longitudinal anastomotic vessel; ISV intersegmental vessel; aISV arterial ISV; vISV venous ISV. Scale bars, 20 μm. Source data are provided as a Source data file.

the ED (Marcksl1a/bΔED) specifically in ECs, the increase in vessel diameter caused by the overexpression of full-length Marcksl1 protein was reduced in most vessels (Fig. 2a–f and Supplementary Fig. 2b, d, e, g and h). We additionally generated dephosphorylated (Marcksl1a-T124A and Marcksl1b-T162A) and phosphomimetic (Marcksl1a-T124D and Marcksl1b-T162D) zebrafish Marcksl1 constructs that contain mutations in putative JNK phosphorylation sites (Fig. 2a, d, and Supplementary Fig. 3). The overexpression of phosphomimetic Marcksl1, particularly Marcksl1a, specifically in ECs also partially reversed

the effect of wildtype Marckl1a/b on vessel diameter (Fig. 2b, c, e, f). These observations therefore suggest that the ED and JNK-mediated phosphorylation of the protein modulate Marcksl1-induced increase in vessel diameter.

We next addressed whether the loss of Marcksl1 function will perturb vessel formation by generating zebrafish harbouring mutations in *marcksl1a* and *marcksl1b* (Supplementary Fig. 4a–d). At 1 and 2 dpf, *marcksl1a*[rk23], *marcksl1b*[rk24] and *marcksl1a*[rk23];*marcksl1b*[rk24] embryos look morphologically normal when compared to wildtype embryos (Supplementary Fig. 4e).

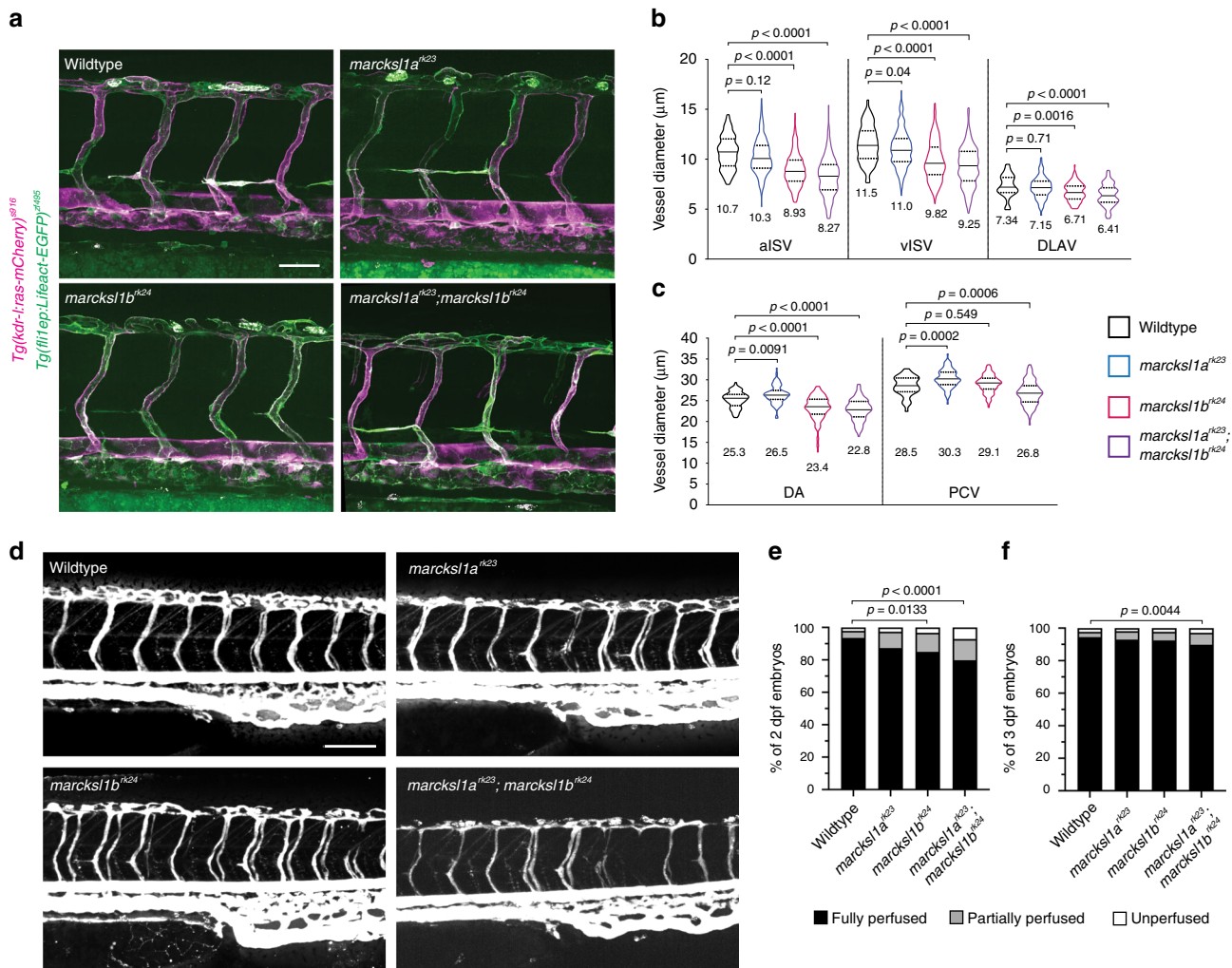

**Fig. 3 Depletion of *Marcksl1* expression alters vessel diameter and delays lumenisation. a** Maximum intensity projection of confocal z-stacks of zebrafish trunk vessels of 2 dpf wildtype, *marcksl1a^rk23^*, *marcksl1b^rk24^* and *marcksl1a^rk23^;marcksl1b^rk24^* embryos in *Tg(fli1ep:Lifeact-EGFP)^zf495^;Tg(kdr-l:ras-mCherry)^s916^* background. **b**, **c** Quantification of aISV, vISV and DLAV (**b**) and DA and PCV (**c**) in wildtype and *marcksl1* mutant embryos (wildtype: n=26 aISVs/27 vISVs from 15 embryos; *marcksl1a^rk23^*: n = 43 aISVs/63 vISVs from 29 embryos; *marcksl1b^rk24^*: n = 79 aISVs/57 vISVs from 39 embryos; *marcksl1a^rk23^*; *marcksl1b^rk24^*: n = 74 aISVs/46 vISVs from 28 embryos). Data are collected from 2 (*marcksl1a^rk23^;marcksl1b^rk24^* embryos), 3 (*marcksl1a^rk23^* and *marcksl1b^rk24^* embryos) and 4 (wildtype embryos) independent experiments. Violin plots represent the entire range of values, dotted lines indicate first and third quartiles, center lines are median. Mean values are indicated. Statistical significance was determined by ordinary one-way ANOVA with Tukey's multiple comparisons test. **d–f** Microangiography in wild type and *marcksl1* mutant embryos. Images are representative of wildtype, *marcksl1a^rk23^*, *marcksl1b^rk24^* and *marcksl1a^rk23^;marcksl1b^rk24^* embryos at 2 dpf. Quantification of lumenisation of ISVs in wildtype and *marcksl1* mutant embryos at 2 dpf (**e**, n = 1472 ISVs from 32 wildtype embryos; n=1058 ISVs from 23 *marcksl1a^rk23^* embryos; n = 1150 ISVs from 25 *marcksl1b^rk24^* embryos; n=1564 ISVs from 34 *marcksl1a^rk23^*; marcksl1b^rk24^ embryos) and at 3 dpf (**f**, n = 1518 ISVs from 33 wildtype embryos; n = 1104 ISVs from 24 *marcksl1a^rk23^* embryos; n = 782 ISVs from 17 *marcksl1b^rk24^* embryos; n = 1748 ISVs from 38 *marcksl1a^rk23^;marcksl1b^rk24^* embryos). ISVs no. 4-26 were analyzed on both sides of the embryo. Data are collected from three independent experiments and analyzed by ordinary one-way ANOVA with Dunnett's multiple comparisons test. DA dorsal aorta; DLAV, dorsal longitudinal anastomotic vessel; ISV intersegmental vessel; aISV arterial ISV; vISV venous ISV; PCV posterior cardinal vein. Scale bars, 50 μm (**a**) and 100 μm (**d**). Source data are provided as a Source data file.

Live imaging of ISV development in *marcksl1a^rk23^;marcksl1b^rk24^* embryos showed normal EC migration and the initiation of lumenisation (Supplementary Movie 1). However, analysis of 2 dpf embryos revealed altered blood vessel diameter in the mutants. *marcksl1a^rk23^* embryos display a small but significant decrease in vISV diameter (Fig. 3a, b) while *marcksl1b^rk24^* and *marcksl1a^rk23^;marcksl1b^rk24^* embryos exhibit significantly decreased diameter in aISVs, vISVs, DLAV and the dorsal aorta (DA, Fig. 3a–c) when compared to wildtype embryos. Additionally, *marcksl1a^rk23^* embryos display a significant enlargement of the DA and posterior cardinal vein (PCV) when compared to wildtype embryos (Fig. 3c). There is therefore a differential response in vessel diameter that is dependent on the vessel type

and the *marcksl1* gene mutated. Nonetheless, the diameter of microvessels (ISVs and DLAV) are consistently decreased, especially by the depletion of *marcksl1b* expression.

By performing microangiography experiment, we additionally discovered a significant decrease in the number of fully perfused ISVs in 2 dpf *marcksl1a^rk23^;marcksl1b^rk24^* embryos when compared to wildtype embryos (Fig. 3d, e). The decrease in fully perfused ISVs was partially alleviated at 3 dpf (Fig. 3f), indicating that although lumen formation occurs in the absence of *marcksl1a* and *marcksl1b*, it is delayed.

In summary, our gain- and loss-of-function experiments indicate a role of Marcksl1 proteins in regulating lumen development and blood vessel diameter.

**Decreased EC number and proliferation in *marcksl1* mutants.** Besides blood flow[12], blood vessel size is determined autonomously by EC number[13,14] and shape[15] as well as non-autonomously by mural cells such as vascular smooth muscle cells and pericytes[16,17]. As the alteration of vessel diameter and morphology is induced by endothelial-specific ectopic expression of Marcksl1a or Marcksl1b and even when the vessel is wrapped by pericytes (Supplementary Fig. 5), we reasoned that the effects of Marcksl1 on vessel diameter is EC autonomous.

We first examined EC number in *marcksl1* mutants by staining 2 dpf embryos with the nucleus marker, DAPI (Supplementary Fig. 6a). In wildtype embryos, the majority of aISVs consist of 3 ECs while most vISVs consist of 4 or 5 ECs. However, in *marcksl1a^rk23*, *marcksl1b^rk24* and *marcksl1a^rk23*;*marcksl1b^rk24* embryos, most aISVs (Supplementary Fig. 6b) and vISVs (Supplementary Fig. 6c) consist of 2 and 2 or 3 ECs, respectively. By performing live-imaging and quantifying the number of cell divisions that occur from 24 to 48 hpf (Supplementary Fig. 6d and e) and staining embryos with the mitotic marker, phosphohistone H3 (Supplementary Fig. 6f), we discovered that ECs of ISVs in *marcksl1a^rk23*;*marcksl1b^rk24* mutants proliferate less frequently compared to wildtype embryos. These results therefore indicate a role of Marcksl1 in regulating EC proliferation to ensure an adequate number of ECs is produced to generate vessels of the correct diameter.

**Marcksl1 regulates EC size to determine vessel diameter.** Mosaic experiments where single ECs with increased Marcksl1a (Fig. 2c) or Marcksl1b (Fig. 2f) expression can induce vessel dilation indicate that Marcksl1 can additionally regulate vessel diameter independent of cell number by regulating cell shape. This is further supported by live imaging that showed fluctuations in ISV diameter without a change in EC number during a 3-hour period (Fig. 4a and Supplementary Movie 2).

By performing in vivo single-cell shape analysis (Fig. 4b), we discovered a significant increase in cell size (Fig. 4c) and cell aspect ratio (Fig. 4d) in Marcksl1b-EGFP-overexpressing EC when compared to control ECs expressing membrane tagged-EGFP (lynEGFP). Conversely, *marcksl1a^rk23*;*marcksl1b^rk24* ECs transplanted into wildtype embryos and ECs of *marcksl1a^rk23*; *marcksl1b^rk24* embryos are, although not statistically significant, smaller than control ECs (Fig. 4c). Importantly, regions of ISVs composed of single *marcksl1a^rk23*;*marcksl1b^rk24* ECs in wildtype embryos have significantly reduced diameter compared to neighbouring segments composed of wildtype ECs (Fig. 4e).

Furthermore, cell shape changes are observed in vitro in human umbilical vein ECs (HUVECs) transfected with plasmid encoding Marcksl1-EGFP. Marcksl1-overexpressing cells displayed significantly increased size (Fig. 4f, g) and abnormal protrusions (cell spikiness, Fig. 4f, h) when compared to control EGFP-transfected cells. Additionally, although Marcksl1-transfected cells are more elongated than control cells at 1 day post transfection (dpt), they failed to elongate further at 2 dpt as observed for control cells (Fig. 4f, i). We additionally developed a plasmid encoding short hairpin RNA to knockdown the expression of *MARCKSL1* in HUVECs (Fig. 4j). As the plasmid also encodes membrane-localised EGFP (EGFP-CAAX), we were able to analyse the effects of reduced *MARCKSL1* expression on single-cell shape. Compared to scrambled shRNA, the transfection of shMARCKSL1 reduced EC area (Fig. 4k) with little or no effect on cell shape (Fig. 4l, m). The finding is further corroborated by siRNA knockdown of *MARCKSL1*, which reduced its expression to 52.9% of siControl cells (Supplementary Fig. 7a) and resulted in a significant decrease in EC area (Supplementary Fig. 7b and c) without a change in cell shape (Supplementary Fig. 7d, e).

Collectively, these experiments demonstrate that the level of Marcksl1 regulates EC size and shape to control vessel diameter.

**Marcksl1 regulates endothelial membrane behaviour during vessel formation.** We discovered that, during the migration and elongation of ISVs at 1 dpf, endothelial Marcksl1a (Fig. 5a and Supplementary Movie 3) or Marcksl1b (Fig. 5b) overexpression resulted in a significant increase in filopodia formation when compared to wildtype ECs. Conversely, the depletion of *marcksl1a* and *marcksl1b* decreased filopodia formation at 1 and 2 dpf (Fig. 5b). This observation concurs with a previous finding that demonstrated MARCKSL1 to induce filopodia formation in neurons[10] and is in line with the role of Marcksl1 as an actin-bundling protein.

Additionally, we detected an unusual membrane behaviour in ECs with elevated Marcksl1 expression when ECs become exposed to blood flow upon lumenisation. During this process, the apical membrane of some ECs expands until the leading edge of the ISV to generate spherical membrane protrusions that resemble blebs on the basal side of the tip cell (Fig. 5c and Supplementary Movie 3). These ectopic blebs are also observed in the DA (Supplementary Movie 4), ISVs (Fig. 5d and Supplementary Movie 4), DLAV (Fig. 5f) and hindbrain vessels (Supplementary Fig. 8) that are composed of ECs with ectopic Marcksl1 expression at 2 and 3 dpf, but are not observed in wildtype blood vessels (Fig. 1b, c, e, f). A measure of membrane blebbing was performed by calculating the ratio between the contour length of a membrane edge and the Euclidean distance between the edge endpoints at each time point of a time-lapse movie (Supplementary Fig. 9). From the resulting time series, the average power spectral density was calculated by Welch's method. Using power spectral density as an indicator of membrane blebbing, we detected a significant increase in blebbing in Marcksl1-overexpressing ECs when compared to wildtype cells at 2 dpf (Fig. 5e).

To further understand the topology of Marcksl1-induced blebs, we filled the lumen of blood vessels with Dextran. Using Marcksl1b-EGFP as a membrane marker, we discovered that newly formed blebs that protrude into the basal side of the cell are filled with luminal content (Fig. 5f and Supplementary Movie 5), demonstrating that the blebs are composed of both apical and basal membranes (Fig. 5g). We therefore refer to these blebs as basal blebs.

In summary, these observations suggest that Marcksl1 may render the apical membrane more deformable by blood flow to promote lumen expansion, and its depletion may impede apical membrane expansion to delay lumen formation, as observed in *marcksl1b^rk24* and *marcksl1a^rk23*;*marcksl1b^rk24* embryos (Fig. 3e, f).

**Reassembly of actomyosin network precedes basal bleb retraction.** Live imaging revealed that Marcksl1-induced basal blebs are highly dynamic, changing shape and size from spherical to more elongated protrusions. Furthermore, while many of the basal blebs retract, some are persistent. As the reassembly of a contractile actomyosin network is required for bleb retraction[18], we next examined the spatiotemporal dynamics of actin in basal blebs. Live imaging shows that there is little actin in the bleb cortex during bleb initiation and initial growth but increases as blebbing continues (Supplementary Movie 6). In some blebs, secondary blebs develop due to anisotropic actin within the primary bleb cortex, with blebs initiating at regions with lower actin density (i in Fig. 6a). When cortical actin reassembly increases to envelop the entire bleb (ii in Fig. 6a), the bleb retracts. However, in cases where actin reassembly fails to increase completely around the bleb cortex, the bleb persists (Fig. 6b) and can even

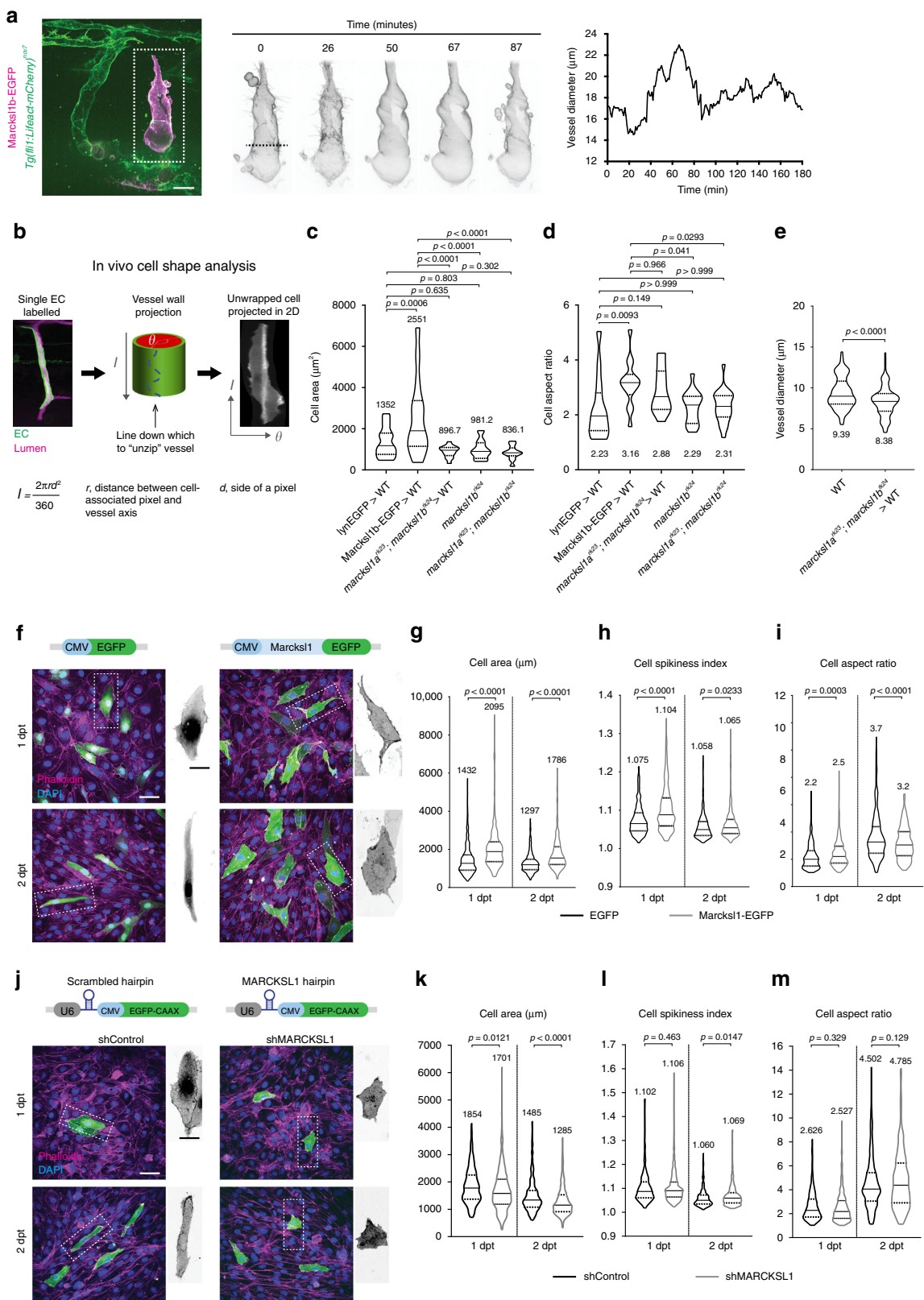

enlarge to form elongated protrusions extending from the vessel wall into the surrounding tissue (Fig. 5d). In addition, we quantified the intensity of actin within the bleb cortex to generate Lifeact intensity kymographs (iii in Fig. 6a) and plots of average Lifeact intensity against bleb area (iv in Fig. 6a), which revealed an inverse relationship between actin assembly and bleb area.

Similarly, live imaging shows that Myl9b strongly accumulates at the base of the bleb as clusters before expanding around the bleb, at which point the bleb retracts (i in Fig. 6c and Supplementary Movie 7). We further quantified non-muscle myosin II assembly in the basal blebs and also found an inverse relationship between Myl9b accumulation and bleb area (ii and iii

**Fig. 4 Marcksl1 expression level controls endothelial cell size and shape. a** Live imaging of an ISV composed of EC with ectopic expression of Marcksl1b from 48 hpf reveals changes in cell shape and fluctuation in vessel diameter. ($n = 5$ embryos, three independent experiments). Dashed line, region of vessel measured. **b–d** Overexpression of Marcksl1 increases EC size while depletion of *marcksl1b* or *marcksl1a* and *marcksl1b* decreases EC size in vivo. Methodology to perform in vivo single-cell shape analysis (**b**). Quantification of cell area (**c**) and cell aspect ratio (**d**) of lynEGFP- or Marcksl1b-EGFP-expressing ECs surrounded by wildtype (WT) cells, *marcksl1*$^{rk23}$;*marcksl1b*$^{rk24}$ ECs transplanted into WT embryo and ECs of *marcksl1b*$^{rk24}$ or *marcksl1*$^{rk23}$; *marcksl1b*$^{rk24}$ embryos at 2 dpf. **e** Diameter of ISVs composed of single *marcksl1*$^{rk23}$;*marcksl1b*$^{rk24}$ EC transplanted into WT embryo at 2 dpf ($n = 10$ vessels, $n = 5$ independent transplantations). **f–m** Overexpression and knockdown of Marcksl1 increases and decreases, respectively, HUVEC size ($n = 3$ independent transfections). Maximum intensity projection of confocal z-stacks of control (EGFP) and Marcksl1-overexpressing (Marcksl1-EGFP) HUVECs 1 and 2 days post transfection (dpt, **f**). Quantification of cell area (**g**), cell spikiness index (**h**) and cell aspect ratio (**i**, EGFP: $n = 423/363$ cells at 1/2 dpt; Marcksl1-EGFP: $n = 358/298$ cells at 1/2 dpt). Maximum intensity projection of confocal z-stacks of control and MARCKSL1 knockdown HUVECs stained for DAPI and phalloidin at 1 and 2 dpt (**j**). Plasmid expressing U6 promoter-driven MARCKSL1 shRNA was used with CMV promoter-driven EGFP-CAAX expression as an internal marker. Plasmid expressing scrambled shRNA was used as a control. Quantification of cell area (**k**), cell spikiness index (**l**) and cell aspect ratio (**m**) after MARCKLS1 knockdown (shControl, $n = 223/278$ at 1/2 dpt; shMARCKSL1, $n = 474/342$ at 1/2 dpt). Statistical significance was assessed by ordinary one-way ANOVA with Sidak's multiple comparisons test (**c**, **d**) and two-tailed unpaired *t*-test (**e**, **g–i**, **k–m**). Violin plots in (**c–e**, **g–i** and **k–m**) represent the entire range of values, dotted lines indicate first and third quartiles, center lines are median. Mean values are indicated. Scale bars, 20 μm (**a**), 50 μm (**f**, **j**) and 25 μm (cropped images in **f**, **j**). Source data are provided as a Source data file.

in Fig. 6c). These observations therefore demonstrate that the retraction of basal blebs is correlated with actomyosin network reassembly at the bleb cortex.

**Marcksl1 decreases cortical resistance to blood flow.** We next addressed why ectopic blebs arise from increased Marcksl1 level and hypothesised that Marcksl1 may decrease membrane-to-cortex attachment, thereby generating basal blebs when subjected to high forces of blood flow. To support this hypothesis, we first tested the effect of weakening the EC cortex of perfused ISVs by UV laser ablation. Local weakening of the cortex resulted in an immediate formation of a bleb at the site of ablation that expanded into the myotome, while adjacent membranes remain unperturbed (Fig. 7a and Supplementary Movie 8). Additionally, inhibition of actin polymerisation by short-term treatment of 2 dpf embryos with 0.3 μg/ml Latrunculin B (Lat. B) resulted in the formation of blebs on the basal membrane of ECs (Fig. 7b, c and Supplementary Movie 9). Together, these findings suggest that the actin cytoskeleton fortifies the cortex of ECs to oppose haemodynamic forces and maintain membrane shape.

We therefore speculated that, if Marcksl1 weakens the EC cortex, membrane blebbing can be normalised by reducing haemodynamic forces acting on the apical surface of ECs in embryos overexpressing Marcksl1. Under the normal dose of tricaine used to anaesthetise embryos (1X tricaine, 160 μg/ml) for live imaging, Marcksl1-induced basal blebs form and retract (Fig. 7d and Supplementary Movie 10). However, upon treatment of the same embryos with 4X tricaine (640 μg/ml) to reduce heart contraction and therefore haemodynamic forces, there is a gradual decrease in blebbing until very few blebs were observed 3 hours later (ii in Fig. 7d). By washing out 4X tricaine and returning the same embryo to 1X tricaine to reinitiate blood flow, inverse blebbing in the apical membrane (i in Fig. 7d) and dilation of some vessels (iii in Fig. 7d) ensued. As *MARCKSL1* mRNA expression is not regulated by shear stress (Supplementary Fig. 10), we conclude that the changes in membrane behaviour by decreased heartbeat is not caused transcriptional changes in *Marcksl1* expression.

In summary, our results demonstrate that excessive levels of Marcksl1 decreases EC resistance to blood flow, leading to blebbing at both apical and basal membranes and widening of blood vessels.

**Marcksl1 promotes the formation of linear actin bundles.** We next sought to uncover the mechanism by which Marcksl1 regulates EC shape and blood vessel diameter. As Marcksl1a and Marcksl1b localise at the plasma membrane, we investigated actin organisation in the EC cortex. High-resolution imaging of

perfused ISVs of 2 dpf embryos revealed that the EC cortex is composed of a meshwork of actin decorated with spots of actin (Fig. 8a). Furthermore, live imaging revealed actin and non-muscle myosin II to be highly dynamic, as demonstrated by the changes in their organisation and fluctuations in their fluorescent intensity over time (Fig. 8c and Supplementary Movie 11). This observation suggests that the cortical actomyosin network is constantly remodelling in perfused blood vessels. As super-resolution imaging in the zebrafish embryo is technically difficult, we investigated whether Marcksl1 alters cortical actin organisation in HUVECs in culture. Like in zebrafish ECs and in a previous finding[19], we also detected a meshwork of actin cytoskeleton at the apical cell cortex (Fig. 8b). In addition, we discovered that Marcksl1-EGFP co-localises with actin (Fig. 8d), supporting previous reports that Marcksl1 binds to and bundles actin[11]. Further analysis revealed that the transfection of Marcksl1-EGFP in HUVECs induced the formation of longer bundles of actin (arrows in Fig. 8e) and a decrease in actin density in the surrounding regions (Fig. 8e, f) without affecting actin bundle width (Fig. 8g) when compared to control cells. Knockdown of *MARCKSL1* using shRNA did not alter actin density nor bundle width at 2dpt (Fig. 8h, i), suggesting that there is a functional redundancy among actin-binding proteins in organising the actin cytoskeleton.

As JNK phosphorylation has been demonstrated to regulate the ability of Marcksl1 to bundle actin filaments[10], we further examined whether JNK-mediated phosphorylation of Marcksl1 affects actin organization by transfecting HUVECs with either dephospho-Marcksl1-EGFP (Marcksl1-AAA-EGFP) or phospho-mimetic Marcksl1-EGFP (Marcksl1-DDD-EGFP) (Fig. 8e). While Marcksl1 phosphomutants did not alter actin density (Fig. 8f), they decreased actin bundle width when compared to control cells (Fig. 8g). We further observed that phosphomimetic Marcksl1 increased bundle width when compared to dephosphorylated Marcksl1, in agreement with previous finding that demonstrated Marcksl1-DDD to induce actin bundling[10]. Analysis of HUVEC morphology (Supplementary Fig. 11a, b) revealed that Marcksl1 phosphomutants increased EC size (compare Fig. 4g and Supplementary Fig. 11c), abnormal protrusions (compare Fig. 4h and Supplementary Fig. 11d) and cell aspect ratio (compare Fig. 4i and Supplementary Fig. 11e) when compared to control EGFP-transfected cells. In addition, Marcksl1 phosphomutants induced a greater number of cell protrusions compared to wildtype Marcksl1, suggesting that a tight regulation of JNK phosphorylation sites is important in controlling membrane protrusions.

Combined with the observation that ectopic expression of Marcksl1 increases filopodia formation in ECs in vivo (Fig. 5a, b), our high-resolution analysis demonstrates that Marcksl1 favours

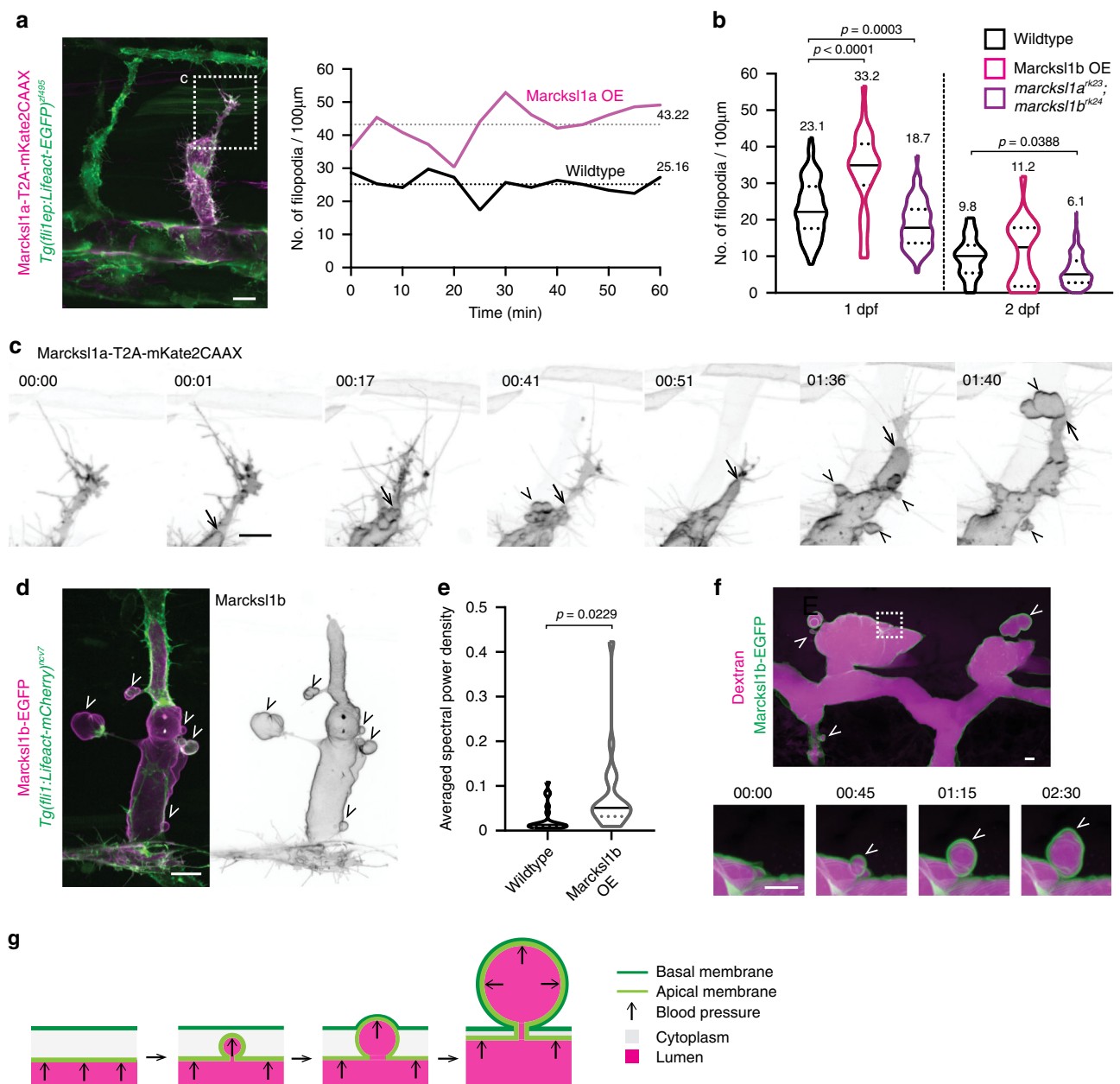

**Fig. 5 Marcksl1 regulates endothelial membrane behaviour during ISV formation. a, b** Marcksl1 regulates filopodia formation. Increased expression of Marcksl1a in ECs (magenta) increases filopodia formation when compared to wildtype EC (green) at 31 hpf (**a**). Dashed box, region of vessel measured. Quantification of wildtype EC or ECs overexpressing Marcksl1b (Marcksl1b OE) or depleted of *marcksl1a* and *marcksl1b* at 1 and 2 dpf (**b**, 1 dpf: wildtype, $n =$ 87 ISVs/ 29 embryos; Marcksl1b OE, $n = 32$ ISVs/19 embryos; *marcksl1a^rk23*;*marcksl1b^rk24*, $n = 67$ ISVs/16 embryos. 2 dpf: wildtype, $n = 29$ ISVs/10 embryos; Marcksl1b OE, $n = 18$ ISVs/15 embryos; *marcksl1a^rk23*;*marcksl1b^rk24*, $n = 126$ ISVs/28 embryos). Statistical significance was assessed by ordinary one-way ANOVA with Sidak's multiple comparisons test. Mean values are indicated. **c** Overexpression of Marcksl1a in ECs causes ectopic membrane blebbing during lumen expansion. Apical membrane (arrow) of an endothelial tip cell expands until the leading edge of the tip cell, at which point ectopic basal blebs (arrowhead) protrude into the surrounding tissue. Movies were taken from 31 hpf. **d** Overexpressing of Marcksl1b in EC (magenta) induces ectopic basal blebs (arrowhead) in the ISV of 54 hpf embryo. **e** Quantification of membrane blebbing shows a significant increase in blebbing in ECs with increased expression of Marcksl1b when compared to wildtype EC of ISVs in 2 dpf *Tg(kdr-l:ras-mCherry)^s916* embryos (wildtype, $n = 21$ cells; Marcksl1b OE, $n = 20$ cells). Statistical significance was assessed by two-tailed unpaired $t$ test with Welch's correction. **f, g** Marcksl1-induced blebs are comprised of apical and basal membranes (green). Newly formed basal blebs (arrowhead) are filled with Dextran (magenta). 00:00; minutes:seconds. Dashed box in **f**, a magnified region. Violin plots in **b**, **e** represent the entire range of values, dotted lines indicate first and third quartiles, center lines are median. Scale bars, 5 μm (**f**) and 10 μm (**a**, **c**, **d**). Source data are provided as a Source data file.

the generation of linear actin bundles in the cell cortex. The increase in the formation of linear actin bundles may concomitantly reorganize the surrounding actin network and create regions with sparse actin within the cortex that may be susceptible to deformations (Fig. 8j).

**Excessive linear actin bundles decrease EC resistance to blood flow.** The excessive formation of linear actin bundles in Marcksl1-overexpressing ECs may consequently perturb the balance of branched and linear actin structures to decrease cortical integrity so that membrane blebs form as a result of pressure

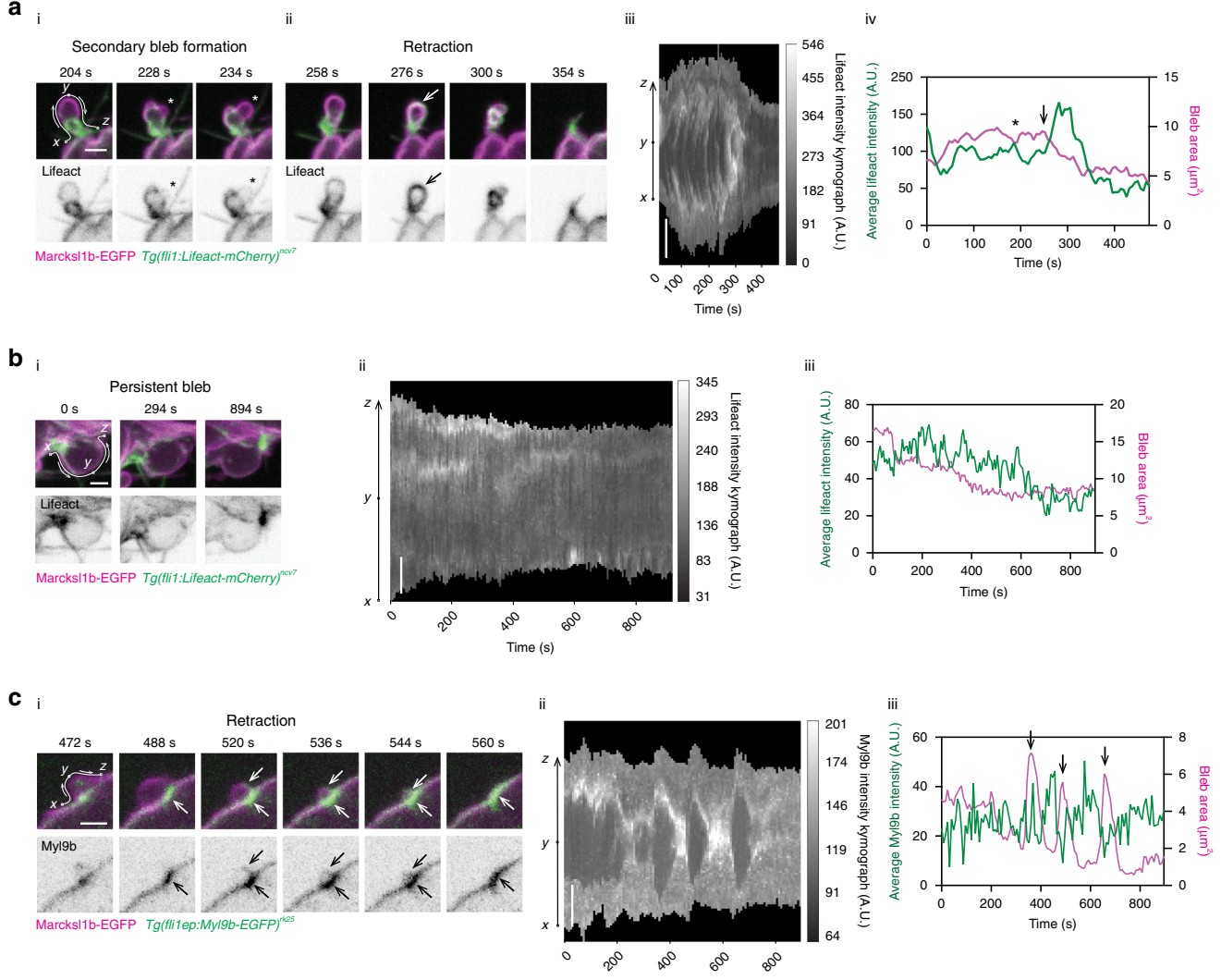

**Fig. 6 Reassembly of actomyosin network at bleb cortex precedes basal bleb retraction.** The spatiotemporal dynamics Lifeact (actin, **a**, **b**; representative images of 17 embryos from 4 independent experiments; secondary bleb formation was observed in 27/100 blebs, retraction in 52/100 blebs and persistent blebs in 27/100 blebs) and Myl9b (non-muscle myosin II, **c** representative image of 21 retracting blebs from 5 embryos from 2 independent experiments) at the cortex of Marcksl1b-induced bleb are depicted in time-lapse images (i and ii in **a**, i in **b**, **c**), kymographs (iii in **a**, ii in **b**, **c**) and as an average intensity over time (iv in **a**, iii in **b**, **c**). Negative correlation between actin (**a**) or non-muscle myosin II (**c**) intensity and bleb area is observed in retracting blebs while little change in actin intensity is observed in blebs that do not retract (**b**). *, secondary bleb. Arrow, increase in actin or non-muscle myosin II assembly. Scale bar, 2 μm.

gradient between the lumen and the cell. To support this hypothesis, we promoted the assembly of linear actin bundles by overexpressing Fascin1a, another actin-bundling protein[20], specifically in ECs and followed the behavior of the EC during lumenisation. Like in ECs with ectopic expression of Marcksl1, we observed the formation of basal blebs in Fascin1a-overexpressing ECs after vessels are exposed to blood flow (Fig. 9a) with some vessels exhibiting wider lumen compared to wildtype EC (Supplementary Movie 12). Additionally, we decreased the formation of branched actin filaments by inhibiting Arp2/3 activity using CK666. Treatment of 2 dpf embryos, when ISVs and DLAV are fully perfused, with 200 μM to 400 μM CK666 for 1 hour resulted in basal blebbing and the deregulation of vessel diameter (Fig. 9b and Supplementary Movie 13; 200 μM, 4 out of 8 embryos; 300 μM, 1 out of 3 embryos; 400 μM, 4 out of 4 embryos), which are rarely observed in wildtype vessels (Fig. 1b, c, e and f and control in Fig. 7c). This experiment therefore

indicates that Arp2/3-dependent branched actin formation is required to resist the deforming forces of blood flow.

In summary, these experiments indicate that a tight balance of linear and branched actin network is necessary for maintaining EC membrane and shape in perfused blood vessels (Fig. 9c). When excessive formation of linear actin bundles is induced at the cell cortex, ECs generate basal blebs and become enlarged because of weakened resistance to haemodynamic forces.

## Discussion

This study underscores the importance of ECs in developing mechanisms to withstand and correct deformations induced by haemodynamic forces. In addition to triggering a repair mechanism to normalize apical inverse membrane blebs during lumenization, we hereby demonstrate that ECs develop an actin-based protective mechanism that prevents blood flow-induced

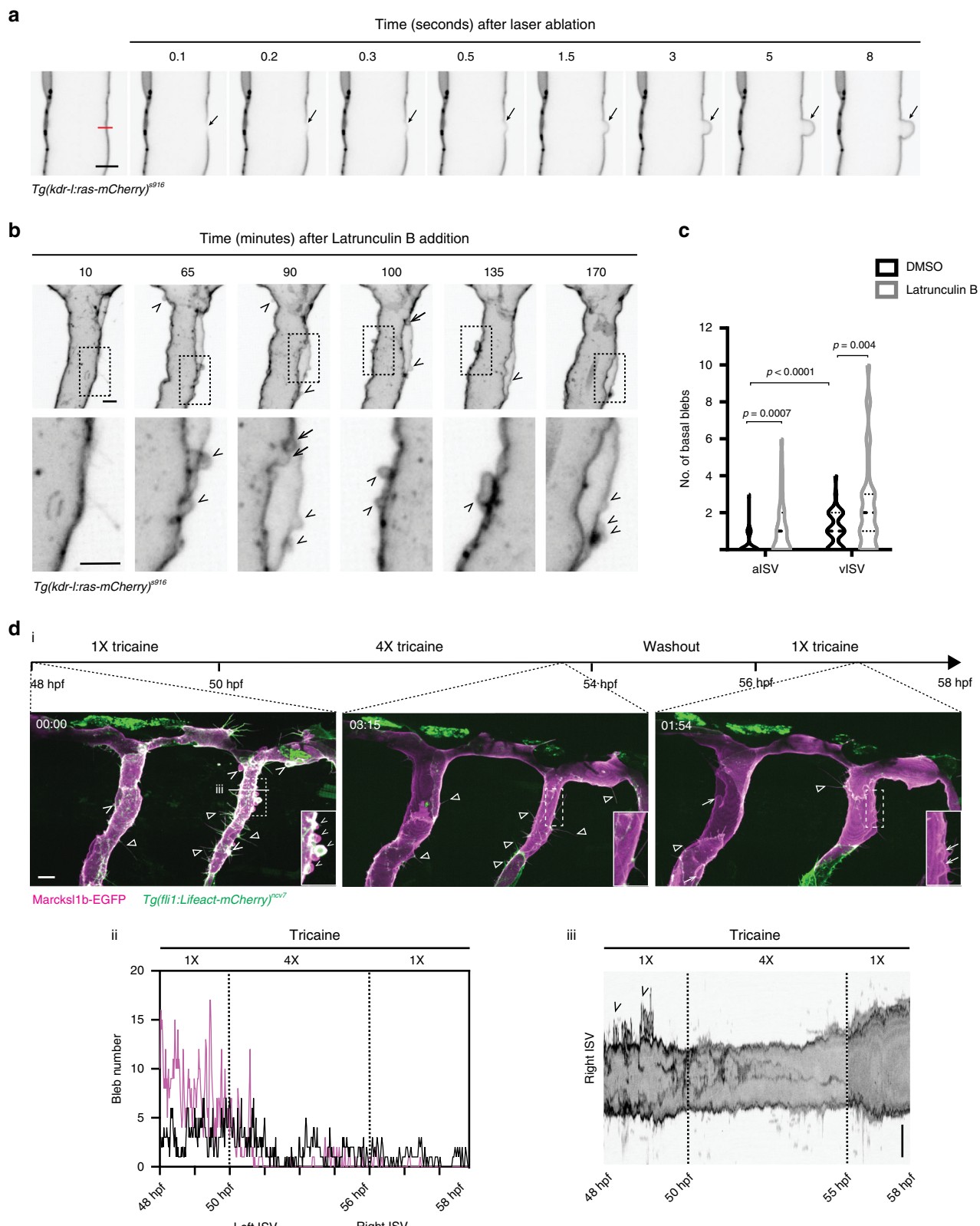

**a** Time (seconds) after laser ablation

0.1    0.2    0.3    0.5    1.5    3    5    8

*Tg(kdr-l:ras-mCherry)^s916*

**b** Time (minutes) after Latrunculin B addition

10    65    90    100    135    170

*Tg(kdr-l:ras-mCherry)^s916*

**c**
□ DMSO
□ Latrunculin B

No. of basal blebs

p = 0.0007
p < 0.0001
p = 0.004

aISV    vISV

**d** i

1X tricaine    4X tricaine    Washout    1X tricaine

48 hpf    50 hpf    54 hpf    56 hpf    58 hpf

00:00    03:15    01:54

iii

Marcksl1b-EGFP    *Tg(fli1:Lifeact-mCherry)^ncv7*

ii
Bleb number

Tricaine
1X    4X    1X

48 hpf    50 hpf    56 hpf    58 hpf

— Left ISV    — Right ISV

iii
Right ISV

Tricaine
1X    4X    1X

48 hpf    50 hpf    55 hpf    58 hpf

deformations to control lumen shape and vessel diameter. This protective mechanism is modulated by the actin-bundling protein, Marcksl1, whose expression level modifies the mechanical properties of the EC cortex to fine-tune EC response to blood flow.

The temporal analysis of actomyosin network assembly reveals that initially, during lumen expansion, actomyosin network

assembly decreases from the posterior to the anterior of the lumen (Fig. 1g). As the actomyosin cytoskeleton regulates cell stiffness[21,22], our observation suggests that differential stiffness within the apical cortex facilitates the expansion of the lumen at the growing front. However, in more developed vessels with robust blood flow, we observe higher levels of actomyosin network at the apical cortex (Fig. 1b, c, e, f). This finding is

**Fig. 7 Decreased blood flow normalizes Marcksl1-induced blebbing. a** Local weakening of EC cortex induces basal bleb. Laser ablation (magenta line) of an arterial ISV was performed in a 3 dpf *Tg(kdr-l:ras-mCherry)^s916* embryo. Bleb formation was observed in 18 out of 35 ISVs (*n* = 15 embryos, 2 independent experiments). Arrow, basal bleb. **b, c** Short-term inhibition of actin polymerisation in 2 dpf *Tg(kdr-l:ras-mCherry)^s916* embryo leads to increased membrane bleb generation at both apical (arrow) and basal (arrowheads) membranes (**b**). Quantification of basal blebs along the length of ISVs in 49–52 hpf *Tg(kdr-l:ras-mCherry)^s916* embryos after treatment with 0.3 μg/ml Latrunculin B for 2.5–3 h (**c**, 0.03% DMSO treatment: aISVs, *n* = 37 ISVs/25 embryos; vISVs, *n* = 39 ISVs/25 embryos; Latrunculin B treatment: aISVs, *n* = 36 ISVs/22 embryos; vISVs, *n* = 39 ISVs/24 embryos). Data were collected from three independent experiments. Violin plots represent the entire range of values, dotted lines indicate first and third quartiles, center lines are median. Statistical significance was determined by two-tailed unpaired *t*-test. **d** Basal blebs (arrowhead) induced by increased expression of Marcksl1b are suppressed upon decreasing blood flow using 4X tricaine (i and ii, reduced blebbing was observed in 7 out of 8 embryos from 6 independent experiments). Reestablishment of normal blood flow (1X tricaine) increases apical membrane blebbing (arrow in i) and vessel diameter (iii). Triangle, filopodia. 00:00, hh:mm after start of treatment. Dashed boxes, magnified region in **b**, **d**. ISV intersegmental vessel; aISV arterial ISV; vISV venous ISV. Scale bars, 5 μm (**a**, **b** and iii in **d**) and 10 μm (i in **d**). Source data are provided as a Source data file.

consistent with a previous report that demonstrated an increase in the density of actin and activated phosphorylated myosin light chain at the EC cortex after chronic exposure to elevated hydrostatic pressure[19]. These observations therefore suggest ECs become stiffer after exposure to blood flow.

The increment of actomyosin assembly to stiffen the cell cortex during development is not the only factor crucial for generating the protective mechanism against blood flow. Importantly, we discovered that cortical integrity additionally depends on actin density and the balance between branched and linear actin which together provide resistive forces to oppose haemodynamic forces to stabilise cell shape in perfused blood vessels. By performing high-resolution analysis of actin organisation at the EC cortex, we discovered that the cortex is composed of a dynamic meshwork of linear and branched actin. When the balance of linear and branched actin is skewed towards linear by the depletion of branched actin formation (by inhibiting Arp2/3 activity, Fig. 9b) or the promotion of linear actin bundles (by endothelial-specific overexpression of Marcksl1, Fig. 4a, and Fascin1a, Fig. 9a), membrane blebbing and altered cell shape arise. The two phenotypes may arise from different degrees of cortical weakening. It has been reported that, in HT1080 fibrosarcoma cells, blebbing is initiated through the intermediate formation of filopodia after inhibiting Arp2/3 complex[23]. Using correlative platinum replica electron microscopy to examine the cytoskeletal architecture of the cell cortex in detail, the authors discovered that bleb initiation was biased toward filopodial bases, where the cytoskeleton is sparser and therefore potentially exhibit local weaknesses in membrane-to-cortex attachment. In line with this, we often detect blebbing in Marcksl1-overexpressing ECs with increased filopodia formation (Supplementary Movie 3). While blebs arise from the local decrease in membrane-to-cortex attachment, the increase in EC size may be a consequence of decreased actin density over a larger surface of the cell. As the actin cytoskeleton is integral in providing cells with mechanical support[24], a reduction in actin density will decrease the ability of ECs to maintain its shape when exposed to blood pressure, which acts perpendicular to the vessel wall, causing cell enlargement and vessel dilation.

During our analysis of Marcksl1 protein localisation, we discovered that EGFP-tagged Marcksl1a and Marcksl1b shows a higher affinity to the apical membrane than the basal membrane of ECs during lumenisation (Fig. 1h, i). Although the membrane localisation of Marcksl1 is mediated by the N-terminal myristoylation domain and the reversible electrostatic interaction between the basic residues in the ED and acidic lipids in the plasma membrane[25], how Marcksl1 is enriched in the apical membrane remains an open question. As MARCKS, which is highly homologous to MARCKSL1, binds strongly to phosphoinositol 4,5-bisphosphate (PIP2) through its ED[26] and PIP2 is enriched in the apical membrane during lumenization (see localization of PLCδ1PH, a PIP2 biosensor, in i and ii in Fig. 1a), we speculate that differences in membrane composition, the

ability of MARCKS/MARCKSL1 to bind to different molecular partners and its phosphorylation status regulate its subcellular localisation and consequently, contribute to localized actin cytoskeleton remodelling. Additionally, the effects of Marcksl1 are regulated by its expression level. RNAseq experiments demonstrate that the expression of *marcksl1a* and *marcksl1b* is highest at 1 dpf, when there is active vessel lumenisation, and decreases at 3 dpf, when lumenization is mostly completed (Supplementary Fig. 1b, c). Given that Marcksl1 renders membranes more deformable, we propose that Marcksl1 may function to create a developmental window that allows flow-dependent apical membrane expansion during lumen formation that subsequently needs to be dampened after completion. However, when the expression of Marcksl1a or Marcksl1b was ectopically elevated after lumen is formed, ECs bleb uncontrollably to destabilise and alter vessel structure. Thus, the localisation and level of Marcksl1 in ECs at different stages of development are important in regulating EC membrane behavior.

Besides regulating the mechanoresponse of ECs to haemodynamic forces, we also demonstrated that Marcksl1 regulates EC proliferation, in agreement with a reported role of Marcksl1 in regulating cell proliferation during axolotl appendage regeneration[27]. In addition, we detected a significant proportion of ISVs with very thin connections with the DA and excessive vessel pruning in 2 dpf *marcksl1a^rk23;marcksl1b^rk24* embryos when compared to control (Supplementary Fig. 12). As junction remodeling and disassembly leads to vessel pruning[28], and the stability of intercellular junctions depends on the assembly of linear actin bundles[29], Marcksl1 may also regulate intercellular junction remodeling during vessel morphogenesis.

A surprising finding that we made during this investigation was that mechanical support provided by pericytes and the myotome are unable to prevent the formation of basal blebs from ECs, demonstrating that a very steep pressure gradient exists between the lumen and the surrounding tissue. This therefore highlights the importance of ECs in generating a cell-autonomous protective mechanism against haemodynamic forces to regulate vessel morphology in other species such as the mouse, where apical membrane blebbing has been suggested to occur during sprouting angiogenesis[9], and in the prevention of vascular anomalies such as aneurysms that form as a result of weakening of the vessel wall. Furthermore, as many tubular organs enclose pressurised fluids or gases, it will be of interest to know whether the balance of branched and linear actin bundles in the cell cortex is also required for resisting luminal pressure to generate stable tubes of uniform size and shape essential for tissue homeostasis.

## Methods

*Zebrafish maintenance and stocks*: Zebrafish (*Danio rerio*) were raised and staged according to established protocols[30].Transgenic lines used were *Tg(kdr-l:ras-mCherry)^s916* [31], *Tg(fli1ep:Lifeact-EGFP)^zf495* [32], *Tg(fli1:myr-EGFP)^ncv2* [33], *Tg(fli1:myr-mCherry)^ncv1* [34], *Tg(fli1:Lifeact-mCherry)^ncv7* [33], *Tg(fli1:GAL4FF)^ubs3* [37], *Tg*

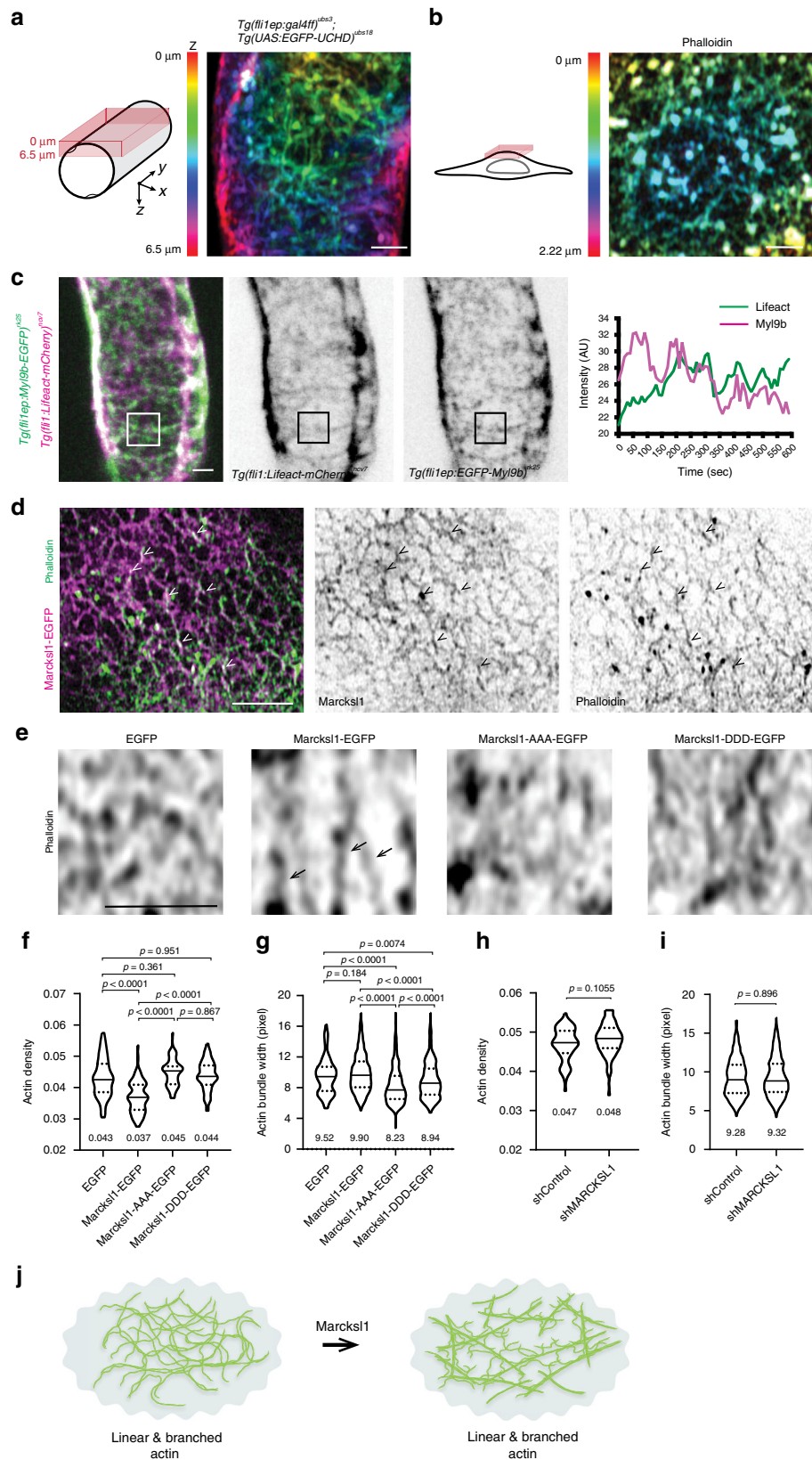

(UAS:EGFP-UCHD)[ubs18] [35], TgBAC(pdgfrb:GFP)[ncv22] [36] and Tg(kdrl:EGFP)[s843] [37]. All animal experiments were approved by the Institutional Animal Care and Use Committee at RIKEN Kobe Branch (IACUC).

Plasmids and oligonucleotides: Zebrafish marcksl1a, marcksl1b and fscn1a were cloned by PCR from cDNA of 1 dpf embryos using NEB® PCR Cloning kit. All expression constructs used in this study were generated via the Gateway® cloning and/or In-Fusion® cloning using plasmids from Tol2Kit[38]. Plasmids with mutated

Marcksl1a and Marcksl1b were generated using Q5® Site-Directed mutagenesis kit (NEB). shRNA containing vectors were constructed by ligation of shRNA sequence between the EcoRI and PmeI sites downstream of the U6 promoter of modified pEGFP-U6 vector[39] (a gift from Zuoshang Xu, Addgene plasmid #19799). Target sequence for human MARCKSL1 is 5′-GTGTGAACGGAACAGATGATG-3′. The scrambled shRNA sequence 5′-GAGTCATGGGTCAGTTATATG-3′ was designed as a mismatched sequence (underlined) of MARCKSL1 shRNA. Vectors containing

**Fig. 8 Marcksl1 regulates actin organisation in the endothelial cell cortex. a–c** EC cortex contains a dynamic meshwork of actomyosin. Maximum intensity projection of Airyscan images of actin in EC from an ISV of a 2 dpf Tg(fli1ep:GAL4FF)^ubs3; Tg(UAS:EGFP-UCHD)^ubs18 embryo (**a**, actin meshwork is observed in 48 out of 63 images from 7 embryos from 5 independent experiments) and the apical cortex of HUVEC stained with Phalloidin (**b**, 21 cells from 4 independent experiments). Time-lapse imaging of actin and non-muscle myosin II in EC of 2 dpf Tg(fli1:Lifeact-mCherry)^ncv7;Tg(fli1ep:EGFP-myl9b)^rk25 embryo (**c**). Lifeact and Myl9b intensity in boxed region was quantified. Dynamic F-actin intensity was observed in 23 ISVs/12 embryos from 5 independent experiments. Dynamic Myl9b intensity was observed in 14 ISVs/6 embryos from 2 independent experiments. **d** Maximum intensity projection of 2 z-slices from the apical cortex of HUVEC expressing Marcksl1-EGFP and stained with Phalloidin reveals co-localisation of Marcksl1 and actin bundles (arrowheads, 23 cells from 5 independent experiments). **e, i** Analysis of actin density and bundle width after overexpression of Marcksl1, Marcksl1-AAA and Marcksl1-DDD (**e–g**) and knockdown of MARCKSL1 (**h, i**) in HUVECs. Single slice Fast Airyscan images of the apical cortex of HUVEC stained with Phalloidin reveal decreased actin density surrounding actin bundles (arrow) in Marcksl1-EGFP-transfected cells 1 day post transfection (**e**). **f–i** Mean values are indicated. Three independent experiments were performed (EGFP, 51 ROIs from 28 cells; Marcksl1-EGFP, 86 ROIs from 29 cells; Marcksl1-AAA-EGFP, 76 ROIs from 28 cells; Marcksl1-DDD-EGFP, 77 ROIs from 32 cells; shControl, 66 ROIs from 24 cells; shMARCKSL1, 74 ROIs from 29 cells). Violin plots represent the entire range of values, dotted lines indicate first and third quartiles, center lines are median. Data was analysed by ordinary one-way ANOVA with Sidak's multiple comparisons test (**f, g**) and two-tailed Mann–Whitney test (**h, i**). **j** Model illustrating Marcksl1 favours the formation of linear actin bundles at EC cortex. Scale bars, 2 μm (**a–e**). Source data are provided as a Source data file.

mouse Marcksl1, Marcksl1-S120A/T148A/T183A (Marcksl1-AAA) and Marcksl1-S120D/T148D/T183D (Marcksl1-DDD) subcloned into pEGFP-N1 (Clontech)[10] were gifts from Eleanor T. Coffey (University of Turku). A detailed information regarding plasmids and primers used in this study can be found in Supplementary Tables 1 and 2, respectively.

*RNA isolation and cDNA synthesis*: Total RNA was isolated using TRI Reagent (Epigenetics) and the Direct-zol^TM RNA MicroPrep kit (Zymo Research) and cDNA was synthesized using the SuperScript III® First-Strand synthesis system (Invitrogen) according to the manufacturer's protocols.

*Mosaic expression of constructs and transgenic zebrafish lines*: Tol2-based expression constructs (5–10 pg) were co-injected into one-cell stage zebrafish embryos with 50 pg of Tol2 transposase mRNA transcribed from NotI-linearized pCS-TP vector (a gift from Koichi Kawakami, National Institute of Genetics, Japan) using the mMESSAGE mMACHINE SP6 kit (Invitrogen). For mosaic expression of transgenes, embryos were analysed at 1 or 2 dpf after injections. To generate Tg(fli1ep:myl9b-EGFP)^rk25 and Tg(fli1ep:EGFP-PLC1δPH)^rk26 lines, injected embryos were raised to adults and screened for founders.

*Generation of zebrafish marcksl1a and marcksl1b mutants*: marcksl1a mutants were generated by CRISPR/Cas9-mediated mutagenesis. A single-guide RNA (sgRNA) targeting the second exon (Supplementary Fig. 4a) was generated by using cloning-independent protocol[40]. Cas9/sgRNA RNP complex was assembled just prior to injection and after 5 min incubation at room temperature 200 pg of Cas9 protein (Invitrogen) and 100 pg of sgRNA were co-injected into one-cell stage Tg(fli1ep:Lifeact-EGFP)^zf495;Tg(kdr-l:ras-mCherry)^s916 embryo. marcksl1b mutants were generated by TALEN-mediated mutagenesis. TALEN targeting the first exon of marcksl1b (Supplementary Fig. 4b) was designed using TAL Effector Nucleotide Target 2.0 (https://tale-nt.cac.cornell.edu/node/add/talen-old) and were assembled via the Golden Gate method[41]. TALEN repeat variable di-residues (RVDs) were cloned into an RCIscript-GoldyTALEN vector (Addgene) and capped mRNAs for each TALENs were in vitro transcribed from SacI-linearized expression plasmids using mMESSAGE mMACHINE T3 kit (Invitrogen). One-cell stage Tg(fli1ep:Lifeact-EGFP)^zf495;Tg(kdr-l:ras-mCherry)^s916 embryos were microinjected with 200 pg of RNA (100 pg of each left and right TALEN mRNAs). F0 founders were identified by outcrossing TALEN or CRISPR/Cas9-injected fish with wildtype fish and screening the offspring for mutations at 2 dpf using T7 endonuclease I (NEB) assay (for TALEN-injected fish) or Sanger sequencing (for CRISPR/Cas9-injected fish). Briefly, genomic DNA was isolated from groups of five embryos using HotSHOT method[42] and corresponding genomic regions were amplified. Mutations were assessed by direct sequencing of purified PCR products. F1 progeny of positive F0 were raised to adulthood and heterozygous carriers for mutations were identified by fin-clipping and routine genotyping PCR analysis using the same primers.

Six mutant alleles in marcksl1a and five mutant alleles in marcksl1b were identified during the screening (Supplementary Fig. 13). Allele rk23 harbours a 2-nt deletion which leads to a frameshift after amino acid 51 and premature stop codon at amino acid 56 after 5 missense amino acids (Supplementary Fig. 4a, c). Allele rk24 harbours a 5-nt deletion which leads to a frameshift after amino acid 13 and premature stop codon at amino acid 17 after 4 missense amino acids (Supplementary Fig. 4b, d). For these reasons, we consider marcksl1a^rk23 and marcksl1b^rk24 to be null alleles and focused our investigations on the analyses of these mutants.

*Live imaging*: Embryos were mounted in 0.8% low-melt agarose (Bio-Rad) in E3 medium containing 0.16 mg/mL Tricaine and 0.003% phenylthiourea. Confocal z-stacks were acquired using an inverted Olympus IX83/Yokogawa CSU-W1 spinning disc confocal microscope equipped with a Zyla 4.2 CMOS camera (Andor). Bright-field images were acquired on a Leica M205FA microscope. Images were processed using Fiji (NIH).

*Laser ablation*: Laser ablations were performed using a 405-nm laser (Andor) and FRAPPA module (Andor) fitted on an inverted Olympus IX83/Yokogawa CSU-W1 confocal microscope with an Olympus UPLSAPO 60x/NA 1.2 water immersion objective. Three sequential laser ablations with a dwell time of 1000 μs each were applied at 51% laser power.

*Microangiography*: Microangiography was performed in wild type, marcksl1a^rk23, marcksl1b^rk24 and marcksl1a^rk23;marcksl1b^rk24 embryos. In all, 2 and 3 dpf embryos were injected with 1–2 nL dextran tetramethyl rhodamine or dextran fluorescein (MW = 2000 kDa, Invitrogen) at 10 mg/mL and imaged immediately. Confocal z-stacks were acquired with an Olympus UPLSAPO 10×/NA 0.4 objective. To cover entire embryo, several regions were imaged and stitched using the Stitching plugin in Fiji[43].

*Cell transplantation*: Groups of 15–25 donor cells from marcksl1a^rk23; marcksl1b^rk24 mutant embryos in Tg(kdr-l:ras-mCherry)^s916 background were collected from a random position in the blastoderm and transplanted to the lateral marginal zone in the blastoderm[44] of wildtype recipient embryos at 4.5–6 hpf. Extraction and transplantation were performed by using a CellTram® 4r Oil microinjector (Eppendorf) with a borosilicate glass capillary GC100-15 (Harvard Apparatus Ltd) crafted with a horizontal micropipette puller P-87 (Sutter Instrument Co.) and a MF-900 microforge (Narishige) to obtain a 'spoon' shape smooth tip. During transplantation embryos were kept in agarose coated petri dishes with 0.5x E2 buffer [7.5 mM NaCl, 0.25 mM KCl, 0.5 mM MgSO₄, 75 μM KH₂PO₄, 0.5 mM CaCl₂, 0.35 mM NaHCO₃ and 50 U/mL of Penicillin-Streptomycin]. At 2 dpf, embryos with mCherry-positive ISVs were selected for microangiography and imaging. A total eight independent transplantations were conducted.

*Chemical treatments*: Latrunculin B (Merck Millipore) was dissolved in DMSO to 1 mg/ml and stored at −20 °C. CK666 (SIGMA) was dissolved in DMSO to 50 mM and stored at 4 °C. All compounds were diluted to the desired concentration in E3 Buffer.

*Transfection, siRNA-mediated protein knockdown and shear stress experiment*: HUVECs (Lonza, C-2519A) were cultured in EGM medium (Lonza) and used until passage 4. For plasmid transfections, cells were seeded in Opti-MEM medium (Gibco) at $5.8 \times 10^4$ cells/cm² on poly-L-Lysine and gelatin-coated coverslips and transfected with 2 μg of plasmid using Lipofectamine 3000 (ThermoFisher Scientific). Two hours after transfection, Opti-MEM medium was changed to EGM medium without antibiotics. For siRNA transfection, HUVECs ($7.9 \times 10^4$ cells/cm²) were transfected with 20 nM siRNA using DharmaFECT1 reagent (Dharmacon). Transfection was repeated after 24 h. Cells were grown for an additional 24 h before total RNA extraction or fixation. ON-TARGETplus human MARCKSL1 siRNA-SMARTpool (Dharmacon) was used to knockdown MARCKSL1. ON-TARGETplus Non-target pool (Dharmacon) was used as control siRNA treatment. Cells were fixed with 4% PFA/PBS, permeabilized with 0.1% Triton X-100 and stained with 14 μM DAPI for nuclei labelling, Alexa 568-phalloidin (1:1000, ThermoFisher Scientific) for visualizing of F-actin, and anti-VE-cadherin antibody (1:100, Cell Signalling) followed by anti-rabbit Alexa 488 secondary antibody (1:1000, ThermoFisher Scientific) to mark EC junctions.

For shear stress experiments, HPAEC (Lonza) were cultured at 1000–1500 cells/mm² on 1% gelatin-coated dish in EGM-2 medium (Lonza) and used before passage 9. The cells were exposed to static or laminar shear stress at 15 dyn/cm² for 0.5, 1 or 6 h using a parallel plate-type apparatus[45,46]. One side of the flow chamber consisted of a 1% gelatin-coated glass plate on which the cultured HPAECs rested, and the other side consisted of a polycarbonate plate. Their flat surfaces were held 200 μm apart with a Teflon gasket. The chamber was provided with an entrance and an exit for the fluid, and the entrance was connected to an upper reservoir with a silicone tube. The exit was open to a lower reservoir. The flow was driven by a roller/tube pump. The fluid (EGM-2 medium) passed from the upper reservoir through the flow chamber into the lower reservoir. The flow rate was monitored with an ultrasonic transit time flow meter (HT107, Transonic Systems) placed at the entrance, and it was controlled by changing the difference in height between the upper reservoir and the exit of the flow chamber. The intensity of the shear stress ($\tau$, dynes/cm2) acting on the EC layer was calculated by using the formula $\tau = 6\mu Q/a^2 b$, where $\mu$ is the viscosity of the perfusate (poise), $Q$ is the flow volume (ml/s), and $a$ and $b$ are the cross-sectional dimensions of the flow path (cm). After shear stress exposure total RNA was isolated for qPCR.

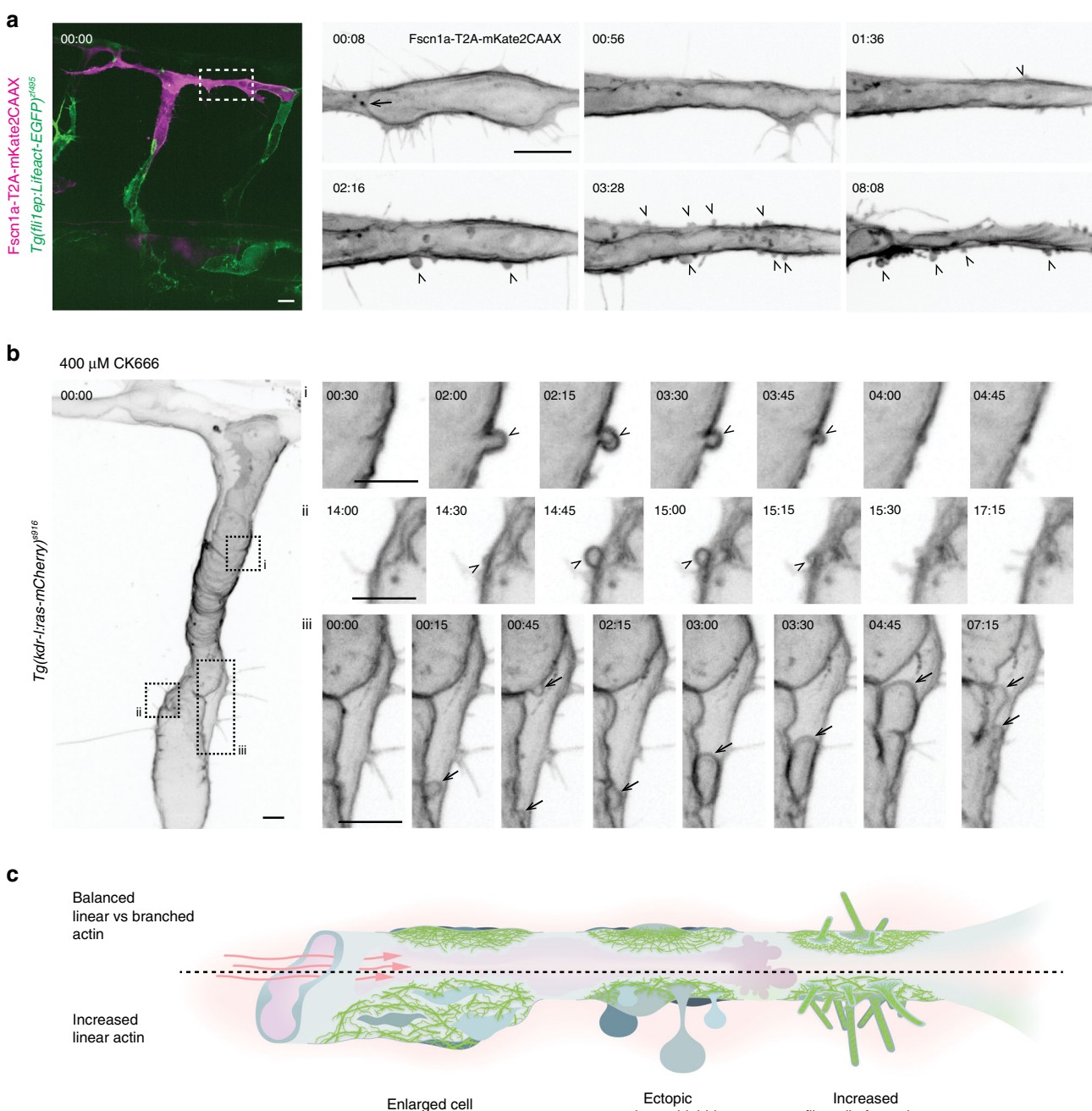

**Fig. 9 Excessive formation of linear actin bundles weakens endothelial cell resistance to blood flow. a** Endothelial-specific ectopic expression of Fascin1a (Fscn1a)-T2A-mKate2CAAX (magenta) at 30 hpf. Box, region of DLAV undergoing lumenisation. Arrow, expanding apical membrane (00:08). Arrowheads, basal blebs (from 02:16). Blebbing was observed in 7 out of 8 ECs with overexpression of Fscn1a (*n* = 6 embryos from two independent experiments). **b** Treatment of a 2 dpf *Tg(kdr-l:ras-mCherry)^s916* embryo with 400 μM CK666 for 1 hour induces basal blebs (i and ii, arrowheads) and apical blebs (iii, arrow; 4 out of 4 embryos from one experiment). Boxes are observed regions. **c** Increase in linear actin bundle formation modifies EC membrane behavior and response to haemodynamic forces. 00:00, hours:minutes. Scale bars, 10 μm (**a**) and 5 μm (**b**).

*Quantitative real-time PCR*: qPCR was performed using 1 μL of the cDNA, generated from 150 ng (HPAECs) or 500 ng (HUVECs) of total RNA, in a 10-μL reaction mix containing 1X Luna Universal qPCR Master Mix (NEB) and 400 nM forward and reverse primers. The reactions were run on ABI Prism 7900HT instrument in quadruplicate. Data were analyzed using the RQ Manager software (Applied Biosystems) and *MARCKSL1* mRNA relative expression was calculated with the $2^{-\Delta\Delta Ct}$ method[47] using *GAPDH* as the reference gene.

*Whole mount in situ hybridization*: Whole mount in situ hybridization was conducted according to standard protocol[48]. Embryos were fixed in 4% PFA at 24, 48 and 72 hpf and permeabilized with 10 μg/mL proteinase K. Hybridization was carried out in buffer [50% formamide, 5x SSC, 50 μg/mL heparin, 500 μg/mL tRNA (Roche) and 0.1% Tween-20] containing 5% sodium dextran sulfate (MW = 500 kDa) at 70 °C overnight. After stringency wash, the specimens were blocked with

2% blocking reagent (Roche) in maleic acid buffer [100 mM maleic acid, 150 mM NaCl, 50 mM MgCl₂ and 0.1% Tween-20, pH 7.5] and incubated overnight with anti-digoxigenin (DIG) antibody, conjugated with alkaline phosphatase (Roche, 1:5000) at 4 °C. Embryos were stained with NBT/BCIP solution (Roche) in staining buffer [50 mM Tris-HCl, 50 mM NaCl, 25 mM MgCl₂, 2% Polyvinyl alcohol and 0.1% Tween-20, pH 9.5]. Embryos were cleared in 70% glycerol overnight and imaged using Leica M205FA microscope. The *marcksl1a* and *marcksl1b* DIG-labeled 1.2 kb long riboprobes were generated from plasmids encoding the full-length cDNAs using MEGAscript SP6 or T7 kits (Invitrogen).

*Single-cell RNA sequencing*: ECs were isolated from *Tg(kdrl:EGFP)^s843* transgenic embryos. Cell sorting was carried out with FACSAriaII Cell Sorter (BD Bioscience). Single-cell suspension was loaded into the 10X Chromium system and cDNA libraries were constructed using Chromium Next GEM Single Cell 3′ GEM, Library

and Gel Bead Kit v2 (10X Genomics) according to manufacturer's protocol. Library was sequenced on the Illumina HiSeq 1500 Sequencer (Illumina, USA). Cell Ranger v2.1 was used in order to de-multiplex raw base call (BCL) files generated by Illumina sequencers into FASTQ files, perform the alignment, barcode counting, and UMI counting. Sequencing reads were mapped to the zebrafish genome assembly (GRCz11, Ensembl release 92). Further analyses were performed using Seurat package[49] in R software (https://www.r-project.org). The expression matrices were first filtered by keeping genes that are expressed in a minimum of 3 cells and cells that expressed a minimum of 200 genes for downstream analysis. Cells were further filtered by selecting for cells that express low mitochondrial content (<7%). The data were then normalized using NormalizedData function which normalizes the gene expression for each cell by the total expression.

*Quantification of blood vessel diameter*: Live *Tg(fli1ep:Lifeact-EGFP)$^{zf495}$;Tg(kdr-l:ras-mCherry)$^{s916}$* transgenic wildtype or *marcksl1* mutant embryos were imaged at 50-52 hpf. Confocal z-stacks were acquired with an Olympus UPLSAPO 40×/NA1.25 silicone oil immersion objective. For DA and PCV diameter calculations, 4–5 measurements were taken between ISVs along the yolk extension (ISVs no. 10-14). For ISV diameter, an average of 6 measurements were made along each ISV (ISVs no. 10-14) between the DA and the DLAV. Measurements were done using Fiji (NIH).

*Quantification of EC number and proliferation*: To count EC number, 52 hpf wildtype, *marcksl1a$^{rk23}$*, *marcksl1b$^{rk24}$* or *marcksl1a$^{rk23}$;marcksl1b$^{rk24}$* embryos in *Tg(kdr-l:ras-mCherry)$^{s916}$* background were fixed in 4% PFA and stained with 3 µM DAPI. Confocal z-stacks were acquired with an Olympus UPLSAPO 40x/NA 1.25 silicone oil immersion objective. Nuclei were counted in each ISV (ISVs no. 9-22) between the DA and the DLAV. To quantify mitotic ECs in ISVs, wildtype or *marcksl1a$^{rk23}$;marcksl1b$^{rk24}$* embryos in *Tg(fli1ep:Lifeact-EGFP)$^{zf495}$;Tg(kdr-l:ras-mCherry)$^{s916}$* background were fixed in 4% PFA at 30, 36 and 48 hpf, permeabilized in 1% DMSO/1% Triton X-100 and stained with anti-phospho H3 antibody (1:250, Merck Millipore) followed by anti-rabbit Alexa 488 secondary antibody (1:1000, ThermoFisher Scientific) and 3 µM DAPI. Confocal z-stacks were acquired with an Olympus UPLSAPO 40x/NA 1.25 silicone oil immersion objective. Mitotic cells (pH3-positive cells) were counted in ISVs on both sides of ISVs (ISVs no. 9-22). ISVs were visualized using mCherry fluorescence. Only one pH3-positive cell per ISV was detected regardless of ISV position. During counting, we considered EC in telophase as a single cell. To count the number of EC divisions, wildtype or *marcksl1a$^{rk23}$;marcksl1b$^{rk24}$* embryos in *Tg(fli1ep:Lifeact-EGFP)$^{zf495}$;Tg(kdr-l:ras-mCherry)$^{s916}$* background were imaged from 24 to 48 hpf at 6 min intervals using an Olympus UPLSAPO 30×/NA 1.05 silicone oil immersion objective. The number of cell divisions in each ISV (ISVs no. 11-15) was quantified.

*Quantification of actomyosin levels in the apical cortex*: Live *Tg(fli1:Lifeact-mCherry)$^{ncv7}$; Tg(fli1ep:EGFP-PLCδ1PH)$^{rk26}$* and *Tg(fli1ep:myl9b-EGFP)$^{rk25}$; Tg(kdr-l:ras-mCherry)$^{s916}$* zebrafish embryos were imaged at 30–34 hpf using an Olympus UPLSAPO 60×/NA 1.2 water immersion objective. The intensity of Lifeact or Myl9b along the apical cortex and in the cytoplasm was measured using Fiji. The ratio of average cortical to cytoplasmic intensity was calculated.

*In vivo cell shape analysis*: Single-cell labelling of ECs in ISVs was achieved by mosaic expression of either *fli1ep:lynEGFP* plasmid in wildtype, *marcksl1b$^{rk24}$* or *marcksl1a$^{rk23}$;marcksl1b$^{rk24}$* mutant embryos or *fli1ep:marcksl1b-EGFP* plasmid in wildtype embryos. At 2 dpf, microangiography was performed and vessels were imaged with an Olympus UPLSAPO 60×/NA 1.2 water immersion objective with optical Z planes interval of 0.25 µm. Cell shape analysis of ECs of ISVs was performed in a semi-automated manner using a custom-written ImageJ script developed in Python, extending the method described in ref. [50]. Images were first crudely cropped to reduce downstream memory requirements and additional metadata (vessel type and embryo identifier) were populated. A separate script was then run to project and unwrap cells from around a vessel onto a 2D surface. Briefly, the cropped images were first rotated based on vessel type to crudely align the vessel axis with the Z direction in an image stack, and the channel of the fluorescent mosaic cell label was selected for projection. To overcome problems with automated vessel segmentation in the blood pool fluorescence channel caused by variable fluorescent background structures and perfusion of dextran-rhodamine outwith the vessel, the vessel axis was identified by manually defining the vessel position approximately every 10 µm through the image stack and then performing a Catmull-Rom spline fit and interpolation. The data was then transformed in order to straighten the vessel axis in three dimensions. Next, the position of maximum intensity in the projection channel was identified along rays at 1° intervals around the vessel axis. This information was used to project fluorescence intensity onto a plane where axes are position along the vessel axis and angular position, and to map distance from the vessel axis at each position on the projected plane. The cell was identified from the projected image using ImageJ's built-in Intermodes method. The arc lengths represented by the sides of each pixel associated with the cell were then calculated ($l = \frac{2\pi r d}{360}$, where $r$ is distance between cell-associated pixel and the vessel axis and $d$ is the side of a pixel) and used to generate measures of cell surface area and aspect ratio.

*In vitro cell shape analysis*: Fixed HUVECs were imaged with an Olympus UPLSAPO 40×/NA 1.25 silicone oil immersion objective. In all, 10–20 images from each cover slip were taken for quantification and analysed in a semi-automated fashion using a custom-written ImageJ script. Multi-channel z-stacks were maximally projected and the channel showing a GFP marker was identified from instrument metadata. ImageJ's built-in Otsu thresholding method was used to separate cells of interest from background; resulting binary images were cleaned by

removing regions smaller than 105 µm² and regions in contact with the boundary of the field of view. A subsequent manual intervention step allowed the user to optionally modify cell regions of interest based on this binary image to separate erroneously merged cells or include cells that were missed by the thresholding operation. The script then calculated areas and perimeters for each cell ROI. In addition, a 'spikiness index' was calculated for each cell based on the ratio of the cell perimeter to the perimeter of the corresponding convex hull, and a value for the aspect ratio of the cell was calculated as the ratio of the major and minor axes of an ellipse fitted to the cell ROI.

*Analysis of membrane blebbing*: Plots relating the membrane blebbing were generated in a semi-automated fashion using a custom-written ImageJ script. For each imaged membrane, the ratio between the contour length of a membrane edge and the Euclidean distance between the edge endpoints was calculated at each time point. From the resulting time series, power spectral densities were calculated by Welch's method[51]. The power spectral density was then averaged over a window of interest, defined empirically as the characteristic time over which blebs formed (0.04–0.06 Hz); these averaged values were compared between experimental (Marksl1b overexpression) and control conditions.

*Analysis of actin and myosin II dynamics in blebs*: Plots relating the dynamics of actin or myosin in blebbing cells were generated in a semi-automated fashion using a custom-written ImageJ script. Multichannel time-lapse z-stacks were first maximally projected along the z dimension. The software then automatically identified the edge of a bleb between two user-defined anchor points at all time points using ImageJ's built-in Otsu thresholding method running on the Marksl1b-EGFP channel as a surrogate for membrane labelling. Following a user quality control step to ensure accurate identification of the bleb edge, the actin/myosin-channel intensity profile along the edge was extracted at each timepoint and plotted against time in a kymograph. At each time point intensity statistics were extracted along with the bleb area, defined here as the z-projected area enclosed by the edge and the line joining the points on either side of the bleb furthest from the bleb tip along an axis perpendicular to the line joining the user-defined anchor points, and average actin/myosin channel intensity and bleb area were plotted against time.

*Quantification of cortical actin density and bundle width*: High resolution images of actin in HUVECs were acquired using a Plan-APOCHROMAT 63x oil objective lens mounted on a Zeiss LSM880 confocal microscope equipped with an Airyscan and GaAsP detector. Approximately 3 µm squared regions within each observation were randomly selected and cropped typically 8 optical sections covering the cortical meshwork out of those regions. Projected images along the Z-axis of cropped stacks were generated by maximum intensity projection and subjected to the following procedure. Since actin filaments within the image were ambiguous and it was impossible to evaluate the whole network, we quantified the number of actin filaments within the limited regions as the density of actin meshwork. This was performed by setting scan lines along X- and Y- axes at every 4 pixels, and pixel values along each scan line were subjected to the Savitzky-Golay filter[52] to calculate the first order derivatives. Actin filaments were then identified by finding zero-crossing points, which are the intensity peaks of the scan line. Counts for peaks were divided by the pixel counts of the scan line (width or height of the image) and the density of the filaments over the image was calculated by taking the average of these values.

To quantify actin bundle width, the centric lines of the actin bundles were generated by finding zero-crossing points of first-order derivatives calculated by the Savitzky-Golay filter and followed by applying the Hilditch thinning algorithm. To eliminate a possibility that branching- or crossing- points of the actin filaments affect measurements, branching points found within the skeletonized line were eliminated and segmented. The orthogonal axis to the overall segment was determined by obtaining the eigenvector, and fluorescent intensities were sampled along the orthogonal axis running the gravity center of the segment with the Lanczos-5 interpolation method. Then, the diameter of a filament was determined by the width of the region that showed larger or equal to half the peak intensity within the line sampled along the orthogonal axis to the overall segment.

*Statistics*: Statistical analysis was performed using Prism 8 (GraphPad Software, Inc.). Sample sizes were not predetermined, the experiments were not randomized, and investigators were not blinded to allocation during experiments and outcome assessment.

**Reporting summary**. Further information on research design is available in the Nature Research Reporting Summary linked to this article.

## Data availability
The data supporting the findings of this study are available within the article and Supplementary Information or from the corresponding author upon reasonable request. A reporting summary for this article is available as a Supplementary Information file. Source data are provided with this paper.

## Code availability
Custom codes generated to analyse in vivo cell shape (https://github.com/dougkelly88/vessel_cell_shape_analysis), in vitro cell shape (https://github.com/dougkelly88/HUVECShapeAnalysis) and membrane blebbing/actin and myosin II dynamics in blebs (https://github.com/dougkelly88/blebbing_analysis) have been deposited in GitHub.

Custom code to analyse actin density and bundle width (ponden) can be obtained from https://dev.bioimageanalysis.jp/.

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

## Acknowledgements

We thank members of the Laboratory for Phyloinformatics, RIKEN Kobe Light Microscopy Facility, RIKEN Aquarium, E. Taniguchi and A. Lagendijk for technical support; E. Coffey for sharing reagents; S. Hayashi and H. Hamada for discussions; H-G. Belting and Y-C. Wang for comments on the manuscript. This work was supported by core funding at RIKEN Center for Biosystems Dynamics Research, the Naito Foundation and the JSPS Grants-in-Aid for Scientific Research (KAKENHI) grant (19K06651) to L-K.P.; JSPS KAKENHI grant (JP19H01022) to N.M.

## Author contributions

Conceptualisation, L-K.P.; experiments, I.K., D.J.K., N.T.C., A.N., J.C., H.N. and L-K.P.; data analysis, I.K., D.J.K., J.C., K.K. and L-K.P.; software development for image

processing and analysis, D.J.K., K.K. and S.O.; resources, N.M. and L-K.P.; writing—original draft, L-K.P.; writing—review and editing, I.K., N.M. and L-K.P.; funding acquisition, L-K.P. and N.M.

## Competing interests

The authors declare no competing interests.
