## [Peer Review File · Nature Communications]

Reviewers' comments:

Reviewer #1 (Remarks to the Author):

The study by Kondrychyn et al. investigates the effects of loss of Marcks1 on the formation of the vasculature in zebrafish embryos. In addition, the authors perform cell culture experiments using HUVECs in order to analyze changes in endothelial cell (EC) shapes upon loss or gain of Marcks1 function. The main conclusions of the paper are that Marcks1 controls EC sizes and thereby blood vessel diameters. They further show that changes in cell shapes are caused by reorganization of the actin cytoskeleton from branched actin to linear actin, partly changing the response to hemodynamic cues. The paper is well written and contains high quality images. Most of the data are thoroughly quantified, allowing strong conclusions about the function of Marcks1. This work will further our understanding of the role of the cytoskeleton on blood vessel diameters. Despite these qualities, the paper has several shortcomings that need to be addressed.

1. The data presented in Figure 1 are not quantified. The statement that there is a “gradient of actomyosin network” (line 87) is not eminent from the data presented. Some additional detail should be provided for non-zebrafish experts about the vessels shown. As a general remark, the numbering of the figures is not ideal (small caps letters and capital letters in the same figure), which makes it sometimes hard to understand what the authors are referring to in the text.

2. In Figure 2, the authors show that the DA and PCV in marcks1a mutants is actually increased in size. Of interest, marcks1b mutants do not show a difference in PCV diameters, while in double mutants PCV size is decreased again. This is in contrast to ISVs. Evidently, marcks1 proteins affect blood vessel diameters differently, depending on the blood vessel investigated. While it is likely beyond the scope of this study to understand the exact reason causing these differences, the authors need to comment on this beyond their current statement that it is “interesting to note” (line 140). Could there be non-EC autonomous effects? Could there be effects on membrane lipid composition? The abstract should be changed in order to accommodate these differences, as the function of marcks1 is not generalizable for all blood vessels (lines 23, 24).

3. The zebrafish in vivo data on cell shapes rely mostly on overexpression results. It would be desirable to complement the single cell analysis in zebrafish overexpressing macks1 with single marcks1 loss of function cells. Are these cells smaller and do ISVs containing these cells show a reduction in diameter? This is particularly important in the light of the finding that ISVs in marcks1 mutants contain fewer endothelial cells. How much do changes in cell numbers vs. changes in cell sized contribute to the observed changes in ISV diameters?

Reviewer #2 (Remarks to the Author):

This is an extremely well-written paper which presents a series of state-of-the-art imaging and genetically-manipulated cell and animal (zebrafish) models to show a strong correlation between the presence of MARCKSL1 and a series of contractile events in the apical cortex of endothelial cells in

the developing embryo which regulate the various facets of vessel growth and lumenization. The studies are well - controlled and data certainly valid, and there is no question that MARCKSL1 plays an integral role in these processes, based on these studies.

An overlying question in this study has to do with MARCKSL1 as the actin binding protein involved in these processes and why this particular kinase substrate plays such an important role in vessel development. Certainly, there are numerous other actin-binding and bundling/unbundling proteins that can do the same job as MARCKSL1, and can do so without the problem of MARCKSL1 phosphorylation, binding to other cellular components, such as calmodulin and PIP2, and membrane insertion and binding via electrostatic interactions with the effector domain. Thus, is there a role in this process for kinases that phosphorylate MARCKSL1, such as protein kinase C or serine/threonine kinases? What is the state of MARCKSL1 phosphorylation as it regulates cortical actomyosin dynamics? How is MARCKSL1 dephosphorylated if indeed it is phosphorylated by kinases during this process, and how do these modifications of MARCKSL1 influence its behavior with regard to actin binding? Clearly, MARCKSL1 needs to be at the apical cell cortex and is required for the contractile events to occur, but, given all the different parameters related to MARCKSL1 function, the paper would be aided invaluablely by a complete discussion of how and why MARCKSL1 actually functions in this process other than actin binding.

Reviewer #3 (Remarks to the Author):

In this manuscript, Kondrychyn et. al. provided a strong comprehensive imaging data on the activation of Marcksl1 resulting in membrane blebbing and vascular formation. They further implied that these deformations in the cell membrane are driven by alterations in cytoskeleton arrangements and important for the initiation of angiogenesis. Genetically manipulated zebrafish are extensively used as a model organism and the species conservation of their phenotype was validated in vitro in HUVEC cells. Although the imaging experiments in the study are quite striking, the manuscript is lacking a mechanistic exploration of Marcksl1 function.

1) The authors claim that Marcksl1 is an important regulator of cell proliferation. However, these conclusions are drawn from observational data alone. The authors should include standard molecular readouts and assays to determine the effect of Marcksl1 on cell proliferation.

2) Although the zebrafish models nicely characterize the effect of Marcksl1 on the endothelium, the pathophysiological significance of these findings are not clear. The use of a pathological stimulation such as hypoxia would be useful. At a minimum, the authors should discuss the translational significance.

3) The authors claim that hemodynamic forces regulate the expression of Marcksl1 based on the administration of triacaine to alter the flow rates. However, this pharmacological approach may have a direct effect of endothelial cells. The authors should include an in vitro shear stress experiment and/or partial artery ligation model to investigate these effects.

4) The species conservation of Marcksl1 is not clear and there is a lack of in vivo evidence to validate the species conservation of the observed phenotype.

5) Most of the supplemental videos are missing the control comparison videos. These should be

provided to orient the reader.

Minor comments:

1) What is the significance of discussing Kugel? This topic seems off topic for the manuscript.

2) The quantification of the conclusions should be provided in figure 1.

3) The authors should expand on how they identified Marcks1. Additionally, several loss of function mutants were explored, the authors should go into more detail into how those mutations were identified and why they were selected.

Response to reviewers' comments

Reviewer #1 (Remarks to the Author):

The study by Kondrychyn et al. investigates the effects of loss of *Marcks11* on the formation of the vasculature in zebrafish embryos. In addition, the authors perform cell culture experiments using HUVECs in order to analyze changes in endothelial cell (EC) shapes upon loss or gain of *Marcks11* function. The main conclusions of the paper are that *Marcks11* controls EC sizes and thereby blood vessel diameters. They further show that changes in cell shapes are caused by reorganization of the actin cytoskeleton from branched actin to linear actin, partly changing the response to hemodynamic cues. The paper is well written and contains high quality images. Most of the data are thoroughly quantified, allowing strong conclusions about the function of *Marcks11*. This work will further our understanding of the role of the cytoskeleton on blood vessel diameters. Despite these qualities, the paper has several shortcomings that need to be addressed.

We thank the reviewer for thoughtful feedback on our findings and the constructive critique. Please find below our response to your concerns.

1. The data presented in Figure 1 are not quantified. The statement that there is a “gradient of actomyosin network” (line 87) is not eminent from the data presented. Some additional detail should be provided for non-zebrafish experts about the vessels shown.

Quantification of actin (Lifeact) and non-myosin II (Myl9b) at the apical cortex at the expanding front (anterior) and the posterior segment of the lumen has now been performed. As shown in Figure 1g, there is an increase in the intensities of apical Lifeact and Myl9b at the posterior region of the lumen when compared to the anterior region of the same lumen, supporting that there is a gradient of actomyosin cytoskeleton in the apical cortex (lines 89 – 91). Furthermore, to further clarify the type of vessels examined, we have annotated the vessels in Figure 1a and d.

As a general remark, the numbering of the figures is not ideal (small caps letters and capital letters in the same figure), which makes it sometimes hard to understand what the authors are referring to in the text.

We apologise for the confusion and have changed the capital letters to lowercase Roman numerals in Figures 1, 6, 7 and 9.

2. In Figure 2, the authors show that the DA and PCV in *marcks11a* mutants is actually increased in size. Of interest, *marcks11b* mutants do not show a difference in PCV diameters, while in double mutants PCV size is decreased again. This is in contrast to ISVs. Evidently, *marcks11* proteins affect blood vessel diameters differently, depending on the blood vessel investigated. While it is likely beyond the scope of this study to understand the exact reason causing these differences, the authors need to comment on this beyond their current statement that it is “interesting to note” (line 140). Could there be non-EC autonomous effects? Could there be effects on membrane lipid composition?

Indeed, there appears to be a differential response of vessel diameter that depends on the vessel type and the *marcks11* gene mutated. The results indicate that the loss of *marcks11b* has a more pronounced effect on decreasing vessel diameter than the loss of *marcks11a*, and its effect is more prominent in microvessels (such as ISVs and DLAV). These differences may be due to where the genes are expressed and at what levels. As *marcks11b* mRNA is

expressed at higher levels than *marcks1a* in ECs and in more ECs (Supplementary Figure 1), the loss of its activity may induce a stronger phenotype than the loss of *marcks1a* and in vessels in which it is expressed. Furthermore, as the perivascular environment (mural cells and basement membrane) contributes to the regulation of vessel diameter and the mutation of *marcks1a* or *marcks1b* occurred in all cell types, we cannot exclude that there is a non-EC autonomous effect on vessel diameter as a result of loss of *marcks1* function in perivascular cells.

We have now included additional statement in lines 143 – 145, commenting on the differential response of vessel diameter that depends on the vessel type and gene mutated.

The abstract should be changed in order to accommodate these differences, as the function of *marcks1* is not generalizable for all blood vessels (lines 23, 24).

As requested, we have changed “blood vessel diameter” to “diameter of microvessels” for accuracy (line 25).

3. The zebrafish in vivo data on cell shapes rely mostly on overexpression results. It would be desirable to complement the single cell analysis in zebrafish overexpressing *marcks1* with single *marcks1* loss of function cells. Are these cells smaller and do ISVs containing these cells show a reduction in diameter?

This is particularly important in the light of the finding that ISVs in *marcks1* mutants contain fewer endothelial cells. How much do changes in cell numbers vs. changes in cell sized contribute to the observed changes in ISV diameters?

The question of whether the changes in ISV diameter are caused by changes in cell number or cell shape is answered by the results obtained from mosaic experiments where single ECs overexpress full length *Marcks1a/b*. Here, an increase in cell area is observed (Figure 4c) and this is accompanied by an increase in vessel diameter (Figure 2b, c, e and f). Furthermore, time-lapse imaging of vessels composed of single *Marcks1b*-overexpressing EC exhibit fluctuations in vessel diameter without a change in cell number (Figure 4a and Figure 7d). We have also expressed mutant *Marcks1a/b* protein lacking the actin-binding effector domain, which perturbs *Marcks1* function, in single ECs and the increase in vessel diameter was no longer observed (Figure 2b, c, e and f). As these experiments are independent of cell number, we reason that it is the effect of *Marcks1* on cell shape, not cell number, that controls ISV diameter.

Although the reviewer is concerned that many of our studies on cell shape rely on the overexpression of *Marcks1*, we want to highlight that we have also performed cell shape analysis on single *marcks1b* mutant ECs and *marcks1a;marcks1b* double mutants ECs (Figure 4c and d), albeit in the mutant background. These experiments demonstrate decreased cell area in ECs lacking *marcks1b* or both *marcks1a* and *marcks1b*; and these mutants have decreased ISV diameter (Figure 3b). Of course, the ideal experiment will be to analyse single mutant ECs in a wildtype background. For this, we have learnt and performed transplantation experiments where cells from *marcks1a^{rk23};marcks1b^{rk24}* zebrafish were transplanted into wildtype zebrafish. Although not statistically significant, transplanted *marcks1a;marcks1b* double mutant ECs are smaller than wildtype ECs and are similar in size as single *marcks1b* and *marcks1a;marcks1b* double mutants ECs in the mutant background (Figure 4c; lines 180 - 182). Additionally, we measured the diameter of the ISVs that are composed of single *marcks1a;marcks1b* double mutant ECs and found a significant reduction in the diameter compared to neighbouring regions composed of wildtype ECs (Figure 4e; lines 182 - 184). With these new experimental data, we hope that the reviewer is now convinced that changes in EC cell size contribute to changes in ISV diameter.

Reviewer #2 (Remarks to the Author):

This is an extremely well-written paper which presents a series of state-of-the-art imaging and genetically-manipulated cell and animal (zebrafish) models to show a strong correlation between the presence of MARCKSL1 and a series of contractile events in the apical cortex of endothelial cells in the developing embryo which regulate the various facets of vessel growth and lumenization. The studies are well-controlled and data certainly valid, and there is no question that MARCKSL1 plays an integral role in these processes, based on these studies.

An overlying question in this study has to do with MARCKSL1 as the actin binding protein involved in these processes and why this particular kinase substrate plays such an important role in vessel development. Certainly, there are numerous other actin-binding and bundling/unbundling proteins that can do the same job as MARCKSL1, and can do so without the problem of MARCKSL1 phosphorylation, binding to other cellular components, such as calmodulin and PIP2, and membrane insertion and binding via electrostatic interactions with the effector domain. Thus, is there a role in this process for kinases that phosphorylate MARCKSL1, such as protein kinase C or serine/threonine kinases? What is the state of MARCKSL1 phosphorylation as it regulates cortical actomyosin dynamics? How is MARCKSL1 dephosphorylated if indeed it is phosphorylated by kinases during this process, and how do these modifications of MARCKSL1 influence its behavior with regard to actin binding? Clearly, MARCKSL1 needs to be at the apical cell cortex and is required for the contractile events to occur, but, given all the different parameters related to MARCKSL1 function, the paper would be aided invaluablely by a complete discussion of **how and why** MARCKSL1 actually functions *in this process* other than actin binding.

We thank the reviewer for his/her generous comments, constructive critique and raising the importance of MARCKSL1 phosphorylation and interactions with other molecules in the regulation of MARCKSL1 function.

Previous work on the phosphorylation status of the proteins of the MARCKS family have mostly been done on MARCKS, a protein that share high protein homology to MARCKSL1. In mammalian MARCKS, it has been demonstrated that the protein is phosphorylated by PKC at four sites in the Effector Domain. The phosphorylation by PKC has several effects including release of MARCKS from the membrane into the cytoplasm so that it can no longer bind to actin at the cortex. The phosphorylation of MARCKS additionally prevents its binding to phosphatidylinositol 4,5-bisphosphate (PIP2) in the membrane and promotes binding Transducer of ErbB2 (TOB2), thereby regulating signalling pathways in the cells. As the Effector Domain is also conserved in MARCKSL1, it is also likely that MARCKSL1 is regulated in the same manner as MARCKS by PKC. Furthermore, it has been demonstrated that mammalian MARCKSL1 is phosphorylated at three residues in the C-terminus by JNK. This phosphorylation enables MARCKSL1 to bundle and stabilise F-actin, increase filopodium number and dynamics and retard migration in neurons.

How the phosphorylation status of MARCKSL1 affects the function of the protein and its regulation on actomyosin dynamics is a very important question. However, as an antibody against phosphorylated MARCKSL1 does not, to our knowledge, exist, it is not possible to detect to what degree Marcksl1 proteins are phosphorylated and where phosphorylated Marcksl1 localizes to in ECs during vascular development in the zebrafish embryo. We are also unable to answer how Marcksl1 is desphosphorylated since this is beyond the scope of our paper; however, we can speculate that PP2A may be a candidate since it has been demonstrated to dephosphorylate MARCKS.

In an attempt to address whether the phosphorylation status of MARCKSL1 is important for regulating actin organization, cell shape and vessel diameter, we have investigated the role of JNK phosphorylation in HUVECs and in the zebrafish.

i) We expressed previously published dephospho-MARCKSL1^{S120A,T148A,T183A} and phosphomimetic MARCKSL1^{S120D,T148D,T183D} (Bjorkblom et al, 2012, Molecular and Cellular Biology) in HUVECs and examined the effects of loss- and gain- of JNK-mediated phosphorylation on actin density, actin bundle width and EC shape.

These experiments show that the phosphomutants did not perturb actin density (Fig. 8e and f) but altered actin bundle width (Fig. 8g) when compared to EGFP-transfected cells (lines 304 – 307). Surprisingly, both phosphomutants decreased actin bundle width when compared to EGFP- and wildtype Marcksl1-transfected cells, with actin bundles in phosphomimetic Marcksl1-transfected cells wider than those in dephospho-Marcksl1-transfected cells. These results suggest that the dynamic regulation of JNK phosphorylation sites (by kinases and phosphatases) modulates actin bundle width.

Analyses of HUVEC shape demonstrate that both phosphomutants increased EC size and cell protrusions (Supplementary Fig. 11 a – d) when compared to control cells (Fig. 4f - g). Interestingly, compared to the overexpression of wildtype Marcksl1, both dephospho-Marcksl1 and phosphomimetic-Marcksl1 induced a much greater increase in cell protrusions (or spikiness; lines 307 - 312). These observations suggest that JNK phosphorylation of Marcksl1 modulates the dynamics of cell protrusions, and further investigation into how the phosphorylation status of Marcksl1 affects actin binding and regulation of cell protrusions is needed to fully understand this process.

ii) We generated zebrafish marcksl1a and marcksl1b constructs that contain mutations at putative JNK phosphorylation sites (dephospho-Marcksl1a-T124A, phosphomimetic Marcksl1a-T124D, dephospho-Marcksl1b-T162A and phosphomimetic Marcksl1b-T162D). These constructs were expressed specifically in ECs in zebrafish embryos in a mosaic manner and the overexpression of Marcksl1a/b phosphomutants on vessel diameter was examined.

Similar to the overexpression of wildtype Marcksl1, the overexpression of Marcksl1a/b phosphomutants resulted in an increase in vessel diameter in microvessels (ISVs and DLAV) when compared to control ECs (Fig. 2 b, c, e and f; lines 125 - 131). This is likely because the phosphomutants still retain the Effector Domain. However, the degree of vessel dilation is reduced when compared to the overexpression of wildtype Marcksl1, suggesting that JNK phosphorylation of Marcksl1a/b also regulates the effect of Marcksl1-mediated increase in vessel diameter. However, how actin binding by Marcksl1 is affected by these constructs is unclear and will require super-resolution microscopy or electron microscopy analysis.

Finally, the question of why MARCKSL1 needs to interact with other cellular component in order to carry out its function is indeed highly relevant and critical. We think that its ability to bind to specific molecules regulates its localization, and therefore function, to certain compartments of the cell. For example, MARCKS has been demonstrated to bind strongly to PIP2, and in ECs, PIP2 is highly enriched in the apical membrane (see i and ii in Fig. 1a). It is therefore tempting to speculate PIP2 enrichment in the apical membrane recruits more Marcksl1 to this compartment to specifically render the apical membrane more deformable during blood vessel lumenisation. We have included this discussion in lines 382 - 391 of the manuscript.

Reviewer #3 (Remarks to the Author):

In this manuscript, Kondrychyn et. al. provided a strong comprehensive imaging data on the activation of Marcksl1 resulting in membrane blebbing and vascular formation. They further

implied that these deformations in the cell membrane are driven by alterations in cytoskeleton arrangements and important for the initiation of angiogenesis. Genetically manipulated zebrafish are extensively used as a model organism and the species conservation of their phenotype was validated in vitro in HUVEC cells. Although the imaging experiments in the study are quite striking, the manuscript is lacking a mechanistic exploration of Marcksl1 function.

We thank the Reviewer #3 for their critique. Please find our response to the concerns below.

1) The authors claim that Marcksl1 is an important regulator of cell proliferation. However, these conclusions are drawn from observational data alone. The authors should include standard molecular readouts and assays to determine the effect of Marcksl1 on cell proliferation.

We have performed phosphohistone H3 immunostaining, which an established method to detect mitosis, at 30, 36 and 48 hpf in wildtype and *marcks1^{rk23};marcks1^{rk24}* embryos. Quantification shows a significant decrease in pH3 positive ECs in ISVs of 36 and 48 hpf *marcks1^{rk23};marcks1^{rk24}* embryos when compared to wildtype embryos. These new data are shown in Supplementary Fig. 6f (lines 165 - 166) and support our previous data where we detected fewer EC division events (Supplementary Fig. 6d and e) in *marcks1^{rk23};marcks1^{rk24}* embryos. We hope that the combination of these two experiments is sufficient to convince the reviewer that Marcksl1 regulates EC proliferation.

2) Although the zebrafish models nicely characterize the effect of Marcksl1 on the endothelium, the pathophysiological significance of these findings are not clear. The use of a pathological stimulation such as hypoxia would be useful. At a minimum, the authors should discuss the translational significance.

We thank the reviewer for raising the pathophysiological significance of our findings. Unfortunately, our lab at present has not established any animal models of pathology and so we have not been able to characterise the effect of Marcksl1 on e.g. hypoxia. However, our findings have implications in vascular pathologies such as aneurysms, which are local dilations of the vessel as a result of weakening of blood vessel wall that can rupture and lead to serious health complications and fatality. Our finding that ECs need to develop a protective mechanism against haemodynamic forces so to maintain vessel shape suggests that, in pathologies such as aneurysm, ECs become dysfunctional and are no longer able to activate this protective mechanism, leading to vascular malformation. This translational significance is discussed lines 415 - 416.

3) The authors claim that hemodynamic forces regulate the expression of Marcksl1 based on the administration of tricaine to alter the flow rates. However, this pharmacological approach may have a direct effect of endothelial cells. The authors should include an in vitro shear stress experiment and/or partial artery ligation model to investigate these effects.

I am afraid that the reviewer misunderstood the experiment in which blood flow was decreased by increasing the dose of tricaine. We did not claim that hemodynamic forces regulate the expression of Marcksl1 after increasing the dose of tricaine. Rather, we claim that high Marcksl1 expression weakens EC integrity to the forces of blood flow such that EC membranes are susceptible to deformations (blebbing) and alteration in cell shape. When hemodynamic forces were decreased by decreasing heart contraction using a high dose of tricaine, Marcksl1b-induced membrane deformations subsided. Thus, *the level of Marcksl1 modulates EC response to hemodynamic forces*.

We have nevertheless performed *in vitro* shear stress experiments where ECs were subjected to static conditions or laminar shear stress at 15 dynes/cm² for 0.5, 1 or 6 hours. qPCR results show that *MARCKSL1* mRNA expression is not significantly altered by shear stress (Supplementary Fig. 10; lines 273 - 276). This finding indicates that the alteration of haemodynamic forces *in vivo* pharmacologically (especially in the short term) did not change the expression of Marcksl1 in ECs and, therefore, the changes in EC membrane and vessel are a result of altered haemodynamic forces.

4) The species conservation of Marcksl1 is not clear and there is a lack of *in vivo* evidence to validate the species conservation of the observed phenotype.

Unfortunately, being a small lab, we are unable to establish other animal models of vascular development. We are therefore unable to address the species conservation of the observed phenotype. However, as apical membrane blebbing has previously been suggested to exist in the mouse retina during the lumenization of angiogenic vessels (Gebala et al., 2016), we speculate that the generation of a protective mechanism against the deforming forces of blood flow will also be required in other species and is discussed in lines 413 – 414 of the manuscript. On this note, we would like to point out that membrane blebbing can only be detected in the presence of blood flow. As such, any other animal models used to examine the regulation of membrane blebbing by Marcksl1 (or other proteins) must possess a functioning heart/blood flow in order to investigate this membrane behaviour.

5) Most of the supplemental videos are missing the control comparison videos. These should be provided to orient the reader.

We apologise for not providing enough control comparison videos to aid the reader in understanding the phenomena shown. We have now added control videos or added in the figure legend to refer to another video or figure for controls.

- Supplementary Video 1 – a control video showing ISV formation of wildtype has already been included.
- Supplementary Video 2 – a control video showing ISV from a 2 dpf wildtype embryo has now been included.
- Supplementary Video 3 – this video already contains an internal control, which is the left ISV (green). It is stated in the figure legend that wildtype ECs are green.
- Supplementary Video 4 – this video already contains internal control, which is the wildtype EC that is labelled green. This is stated in the figure legend. We have additionally included in the figure legend to refer to the control in Supplementary video 2 for further control.
- Supplementary Video 5 – a control video showing DLAV with dextran in lumen of a wildtype embryos has now been included.
- Supplementary Video 6 – the control video for actin dynamics in a control embryo is in Supplementary Video 11. We have now added a sentence in the figure legend to refer the reader to Supplementary Video 11.
- Supplementary Video 7 – a control video showing Myl9b-EGFP dynamics in WT ECs is now included.
- Supplementary Video 8 – the control for this experiment are regions of the vessel wall that are not ablated by laser. For clarity, we have added an asterisk (*) to denote the site of ablation and added a new sentence “surrounding membranes not ablated by laser remain unperturbed” in the figure legend.
- Supplementary Video 9 – a video of DMSO-treated ISV has now been included.

- Supplementary Video 10 – the purpose of this experiment is to observe the effects of different blood flow rates (1x tricaine, 4x tricaine, washout followed by 1x tricaine) in the same embryo. We do not think a control video is necessary.
- Supplementary Video 11 – This is a control video showing actomyosin dynamics in a wildtype embryo.
- Supplementary Video 12 – There is already an internal control - wildtype ECs in the imaged embryo - which are the cells in green. We have now indicated that cells in green are wildtype ECs in the figure legend.
- Supplementary Video 13 – A control video showing DMSO-treated ISVs has now been included.

Minor comments:

1) What is the significance of discussing Kugel? This topic seems off topic for the manuscript.

The discussion about Kugeln was to highlight the plasticity in EC shape. We have now removed this section from the manuscript.

2) The quantification of the conclusions should be provided in figure 1.

Quantification of the intensity of apical actin and myosin II at the anterior and posterior regions of expanding lumen has now been performed and is presented in Fig. 1g. We have additionally represented the intensity levels of *Marcks1a*-EGFP and *Marcks1b*-EGFP as fire LUT to illustrate higher intensity in the apical membrane (Fig. 1 h and i).

3) The authors should expand on how they identified *Marcks1*. Additionally, several loss of function mutants were explored, the authors should go into more detail into how those mutations were identified and why they were selected.

Our lab has a long-standing interest understanding how the actin cytoskeleton regulates blood vessel morphogenesis and is therefore interested in molecular regulators of actin cytoskeleton. During the search for actin binding proteins that are expressed in blood vessels in ZFIN (The Zebrafish Information Network), we found that *Marcks1a* is expressed in blood vessels during zebrafish development. As little was known about its role in vascular development, we decided to pursue the study of this protein. As we do not think that this information contributes much to the message of our experimental findings, we did not include it in the manuscript.

More detailed information on how mutated alleles of *marcks1a* and *marcks1b* were identified and selected for further studies have now been included in “Generation of zebrafish *marcks1a* and *marcks1b* mutants” in Methods (lines 712 – 718) and Supplementary Fig. 13.

REVIEWERS' COMMENTS

Reviewer #1 (Remarks to the Author):

The authors have greatly improved the manuscript and addressed all of my concerns.

Reviewer #2 (Remarks to the Author):

The authors have responded appropriately to my previous questions, concerns and suggestions and this manuscript appears appropriate for publication.

Reviewer #3 (Remarks to the Author):

The authors have appropriately addressed my critics. I have no more questions.

Response to the Reviewers' comments

Reviewer #1 (Remarks to the Author):

The authors have greatly improved the manuscript and addressed all of my concerns.

We are delighted that we were able to meet all of the reviewer's concerns.

Reviewer #2 (Remarks to the Author):

The authors have responded appropriately to my previous questions, concerns and suggestions and this manuscript appears appropriate for publication.

We are glad that Reviewer #2 is satisfied with our response to his/her questions, concerns and suggestions.

Reviewer #3 (Remarks to the Author):

The authors have appropriately addressed my critics. I have no more questions.

We are glad that we were able to meet all of the reviewer's critics.